# External globus pallidus input to the dorsal striatum regulates habitual seeking behavior in male mice

Matthew Baker [1,7], Seungwoo Kang [1,2,7], Sa-Ik Hong [1], Minryung Song [3], Minsu Abel Yang [3], Lee Peyton [1], Hesham Essa [1], Sang Wan Lee[3,4] & Doo-Sup Choi [1,5,6] ✉

The external globus pallidus (GPe) coordinates action-selection through GABAergic projections throughout the basal ganglia. GPe arkypallidal (arky) neurons project exclusively to the dorsal striatum, which regulates goal-directed and habitual seeking. However, the role of GPe arky neurons in reward-seeking remains unknown. Here, we identified that a majority of arky neurons target the dorsolateral striatum (DLS). Using fiber photometry, we found that arky activities were higher during random interval (RI; habit) compared to random ratio (RR; goal) operant conditioning. Support vector machine analysis demonstrated that arky neuron activities have sufficient information to distinguish between RR and RI behavior. Genetic ablation of this arky$^{GPe→DLS}$ circuit facilitated a shift from goal-directed to habitual behavior. Conversely, chemogenetic activation globally reduced seeking behaviors, which was blocked by systemic D1R agonism. Our findings reveal a role of this arky$^{GPe→DLS}$ circuit in constraining habitual seeking in male mice, which is relevant to addictive behaviors and other compulsive disorders.

The external globus pallidus (GPe) has often been thought of as a relay nucleus in the indirect pathway of the basal ganglia, receiving input from the dorsal striatum and projecting to downstream output targets[1–3]. The importance of the globus pallidus in motor function and clinical relevance in movement disorders has long been appreciated[4–10]. More recently, the GPe has also been implicated in non-motor functions such as decision-making and reward-seeking behaviors by coordinating output from the dorsomedial (DMS) and dorsolateral (DLS) striatum, which are known to regulate goal-directed and habitual seeking, and compulsive disorders such as addiction[6,11–14].

This is further supported by cell- and circuit-specific characterization of GABAergic GPe neurons showing distinct projections throughout the basal ganglia[6,15]. Evidently, the GPe is subdivided into primarily two types of GABAergic projection neurons, prototypic and arkypallidal (arky)[15–18]. Prototypic neurons represent approximately two-thirds of GPe neurons and innervate the downstream subthalamic nucleus (STN) and other output nuclei[16,18]. In contrast, arky neurons comprise approximately one-quarter of GPe neurons and project back to the dorsal striatum[16,18]. While parvalbumin (PV) is the primary cellular marker for the prototypic neurons, neuronal PAS domain protein 1 (NPAS1) and forkhead box p2 (FOXP2) are two main molecular markers defining arky neurons[15,16,18,19]. Interestingly, increased arky activities have been shown to reduce dorsal striatum neural activities and inhibit dorsal striatum-dependent motor behaviors[20,21]. In addition,

[1]Department of Molecular Pharmacology and Experimental Therapeutics, Mayo Clinic, Rochester, MN 55905, USA. [2]Department of Pharmacology and Toxicology, Medical College of Georgia, Augusta University, Augusta, GA 30912, USA. [3]Department of Brain and Cognitive Sciences, Korea Advanced Institute of Science and Technology (KAIST), Daejeon 34141, Republic of Korea. [4]Department of Bio and Brain Engineering, Korea Advanced Institute of Science and Technology (KAIST), Daejeon 34141, Republic of Korea. [5]Department of Psychiatry and Psychology, Mayo Clinic College of Medicine and Science, Rochester, MN 55905, USA. [6]Neuroscience Program, Mayo Clinic College of Medicine and Science, Rochester, MN 55905, USA. [7]These authors contributed equally: Matthew Baker, Seungwoo Kang. ✉e-mail: choids@mayo.edu

neural activity changes in the DMS and DLS have been shown to underly the transition between goal-directed and habitual reward seeking[11,13,22–26]. However, whether GPe arky neurons also control behavioral inhibition in reward-seeking behaviors through this feedback circuit to the dorsal striatum has not been studied.

In the present study, we provide a role of GPe arky neurons in goal-directed and habitual seeking. Using fiber photometry $Ca^{2+}$ recordings, computational modeling, genetic ablation, chemogenetic, and behavioral approaches, we revealed how the arky$^{GPe \to DLS}$ circuit regulates action-selection and inhibition.

## Results
### Arkypallidal neurons primarily project to the DLS
To determine if GPe arky neurons showed preferential projection to the DMS or DLS, we injected an anterograde virus [AAV5-CaMKII(1.3).eYFP.WPRE.hGH] into the GPe. We then compared the relative fluorescence of synaptic terminals in the DMS or DLS and normalized fluorescence values according to the GPe injection site (Fig. 1a). Interestingly, GPe arky neurons preferentially projected to the DLS compared to the DMS (one-way ANOVA, $p < 0.05$; Fig. 1b). Next, we injected retrobeads into the DMS or DLS and examined the retrograde signal in the GPe (Fig. 1c). Cell counts indicated a significantly higher number of cell bodies in the DLS compared to the DMS injection group (unpaired $t$ test, $p < 0.05$; Fig. 1d). Next, to characterize the cellular markers of DLS-projecting cells in the GPe we injected an mCherry-expressing retrograde virus [AAV-Ef1a-mCherry-IRES-Cre] into the DLS (Supplementary Fig. 1a). Then we imaged and quantified the overlap of GPe cell bodies with either a FOXP2 or PV antibody (Supplementary Fig. 1b). Consistent with previous estimates, we found that most of the DLS-projecting neurons in the GPe expressed FOXP2 (70.9%), but not PV (3.1%; 26% other; Supplementary Fig. 1c)[15–17].

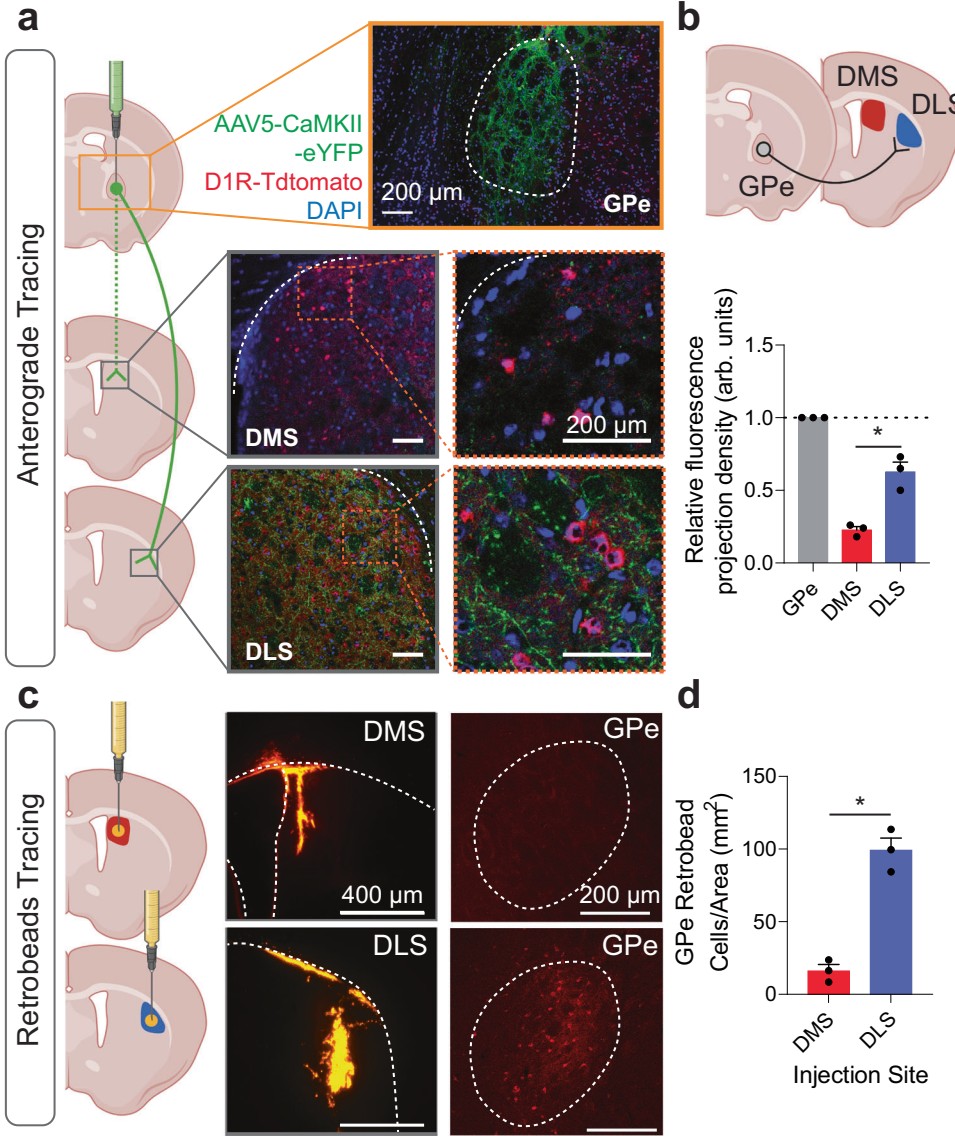

**Fig. 1 | Anterograde and retrograde tracing of external globus pallidus (GPe) projections. a** Injection schematic of AAV5-CaMKII-eYFP virus into the GPe in D1R-TdTomato mice and projection densities in the dorsomedial striatum (DMS) and dorsolateral striatum (DLS). Scale: 200 μm. Anterograde virus (Green), D1R (Red), DAPI (Blue). **b** A schematic circuit showing a majority of GPe projections target the DLS, compared to the DMS ($p = 0.0012$). $n = 3$ images/group. $F_{(2,6)} = 87.60$, $p < 0.0001$. **c** Retrobead tracing of DMS and DLS afferents and representative IHC images. Scale: 400 μm (left), 200 μm (right). **d** DLS retrobead injections showed more GPe projection cells compared to DMS. $n = 3$ images/group. One-way ANOVA with Tukey's posthoc tests were used for (**c**). Two-tailed $t$ tests were used for (**d**). *$p < 0.05$. Data represent mean ± SEM. See Supplementary Table 2 for full statistical information. Source data are provided as a Source Data file. (**a**–**c**) were created with BioRender.com.

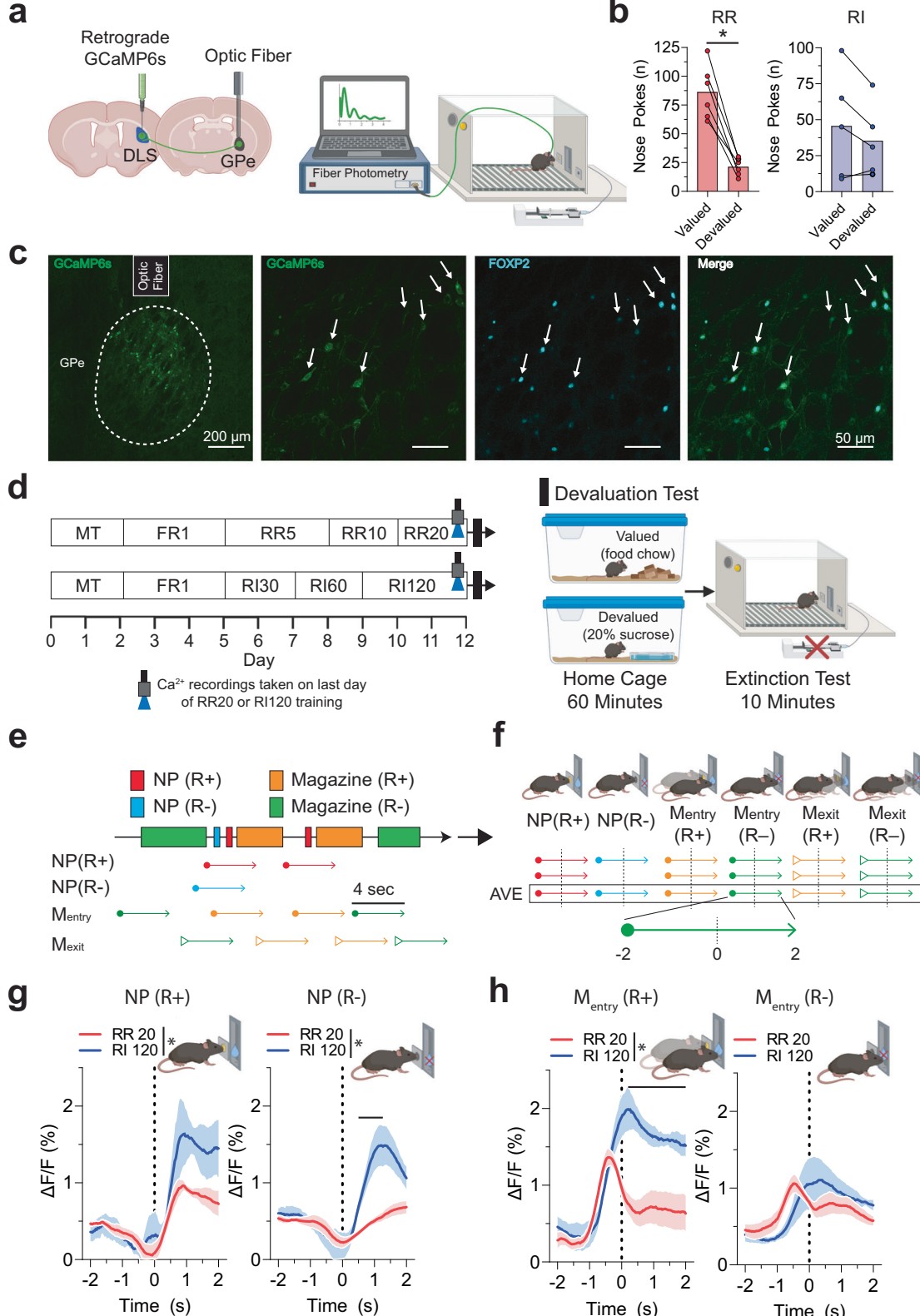

## Mice exhibit goal-directed or habitual behaviors through training on random operant schedules

Based on previous findings that GPe arky neurons are important for regulating dorsal striatum-dependent behaviors[20,21], we sought to determine the temporal dynamics of arky neurons during goal-directed and habitual behavior for a 20% sucrose reward using the genetically encoded calcium-sensitive fluorescent proteins, GCaMP6[27].

A retrograde virus expressing GCaMP6s [AAV-hSyn1-GCaMP6s-P2A-nls-dTomato] was injected into the DLS (Fig. 2a, c), and we recorded intracellular Ca²⁺ signal in the GPe using fiber photometry during the last training sessions of random ratio (RR; goal) and random interval (RI; habit) schedules, where the operant behaviors were presumably sufficiently learned (Fig. 2a, d). During magazine training (MT), both groups of mice showed reduced latency to the magazine from the first

**Fig. 2 | Fiber photometry Ca²⁺ signal of external globus pallidus (GPe) arky-pallidal neurons during random ratio (RR) and interval (RI) operant con-ditioning. a** Schematic of retrograde GCaMP6s virus injection into the DLS, fiber placement over the GPe, and Ca²⁺ recordings in the operant chamber. **b** RR-trained mice ($n = 6$) showed a significant decrease in nose pokes during extinction testing for the devalued state, suggesting of goal-directed behavior ($p = 0.03$). RI-trained mice ($n = 5$) showed no differences between valued and devalued states ($p = 0.62$), suggesting of habitual behavior. **c** IHC images showing representative GCaMP6s expression in the GPe, optic fiber placement, and coexpression with arkypallidal cellular marker FOXP2 from $n = 3$ mice. Scale: 200 µm (left), 50 µm (rest). **d** Operant RR and RI behavior schedules to establish goal-directed and habitual seeking, respectively. Ca²⁺ signal recordings were collected on the last day of training, fol-lowed by evaluation testing to determine reward-outcome valuation testing. **e** Behavioral and arkypallidal neuron Ca²⁺ signal alignment in RR and RI-trained mice surrounding rewarded and unrewarded nose poke. **f** Ca²⁺ signal was recorded 2 s before and after each behavioral event. Solid lines indicate mean Ca²⁺ signal for rewarded (R+) and unrewarded (R−) nose poke (NP). **g** Ca²⁺ signaling in the nose

poke. For the rewarded nose poke (NP+), $F_{(1,7)} = 5.73$, $p = 0.048$ for group (RR vs RI), $F_{(120,840)} = 11.40$, $p < 0.0001$ for time, $F_{(120,840)} = 1.62$, $p = 0.0001$ for interaction. For the non-rewarded nose poke (NP-), $F_{(1,7)} = 12.34$, $p = 0.0098$ for group (RR vs RI), $F_{(120,840)} = 12.52$, $p < 0.0001$ for time, $F_{(120,840)} = 5.52$, $p < 0.0001$ for interaction. **h** Ca²⁺ signaling in the magazine entry. For the rewarded magazine entry [M$_{entry}$ (R+)], $F_{(1,7)} = 25.84$, $p = 0.0014$ for group (RR vs RI), $F_{(120,840)} = 14.48$, $p < 0.0001$ for time, $F_{(120,840)} = 5.73$, $p < 0.0001$ for interaction. For the non-rewarded magazine entry [M$_{entry}$ (R−)], $F_{(1,7)} = 0.05$, $p = 0.83$ for group (RR vs RI), $F_{(120,840)} = 5.60$, $p < 0.0001$ for time, $F_{(120,840)} = 1.74$, $p < 0.0001$ for interaction. Wilcoxon test was used for (**b**). Two-way repeated measures ANOVA with Tukey's posthoc tests were used for (**g, h**). $n = 5$ (RR) and 4 (RI) mice. The solid line indicates the time range of significant posthoc difference between RR and RI Ca²⁺ traces. *$p < 0.05$. Data represent mean ± SEM. See Supplementary Table 2 for full statistical information and Supplementary Table 3 for specific significant time ranges of fiber photometry. Source data are provided as a Source Data file. (**a**, **d**, and **f**–**h**) were created with BioRender.com.

day to the last day. Both RR and RI groups showed increased nose poke rates across sessions and an increased likelihood to choose the active nose-poke hole compared to the inactive hole (Two-way RM ANOVA, $p < 0.05$; Supplementary Fig. 2b–d). In the devaluation test, which compares nose poking between the valued and devalued conditions, RR-trained mice showed a reduction in nose poking in the devalued conditions, indicating goal-directed reward-seeking (Wilcoxon test, $p < 0.05$; Fig. 2b). RI-trained mice did not show a decrease in nose poking in the devalued condition, indicating habitual responding ($p > 0.05$; Fig. 2b).

## Arkypallidal neurons exhibit increased Ca²⁺ signaling during habitual (RI) seeking

After confirming that mice showed goal-directed (RR) or habitual (RI) seeking in the reward-devaluation task, we examined GPe arky neural activities surrounding the six behavioral events: rewarded nose-poke (NP R+), unrewarded nose-poke (NP R−), rewarded magazine entry (M$_{entry}$ R+), unrewarded magazine entry (M$_{entry}$ R−), rewarded maga-zine exit (M$_{exit}$ R+), and unrewarded magazine exit (M$_{exit}$ R−). We aligned the Ca²⁺ signal data 2 s prior and 2 s following each behavioral event (4 s total, 120 frames; Fig. 2e, f). Mice showed increased GPe arky neuron activities during the RI120 task compared to the RR20 task for rewarded and unrewarded nose poke (Two-way RM ANOVA, $p < 0.05$; Fig. 2g), and rewarded magazine entry ($p < 0.05$; Fig. 2h). GPe arky Ca²⁺ signal was significantly higher in the RR20 task for rewarded magazine exit compared to RI 120 ($p < 0.05$; Supplementary Fig. 2e). However, no effect of operant schedule for unrewarded magazine entry or unre-warded magazine exit was observed ($p > 0.05$; Fig. 2h, Supplementary Fig. 2e). Specific time ranges for significant *post hoc* comparisons between operant schedules are presented in Supplementary Table 3.

Additionally, we examined whether the temporal cellular activities at the time of action selection were stable or changed across the duration of an operant session. We used a regression analysis to compare the relationship between the degree of change in Ca²⁺ signal surrounding each of the specific behavioral events with the progres-sion of the trial. Due to the high variability in the total number of behavioral events across individuals and operant schedules, we transformed each trial into 10 blocks, each representing 10% increments of that behavioral event for the session. GPe arky Ca²⁺ signal for RI mice is progressively increased in activity change across the duration of a trial for rewarded nose poke, unrewarded nose poke, rewarded magazine entry, and unrewarded magazine entry (linear regression, $p < 0.05$; Supplementary Fig. 3a–d). During the RR task, arky activities are only increased across the trial duration for rewarded nose poke and rewarded magazine entry ($p < 0.05$; Supplementary Fig. 3a, c). Overall, arky Ca²⁺ signal was increased across trial duration at a greater rate for RI compared to RR for unrewarded nose poke and rewarded magazine entry ($p < 0.05$; Supplementary Fig. 3b, c).

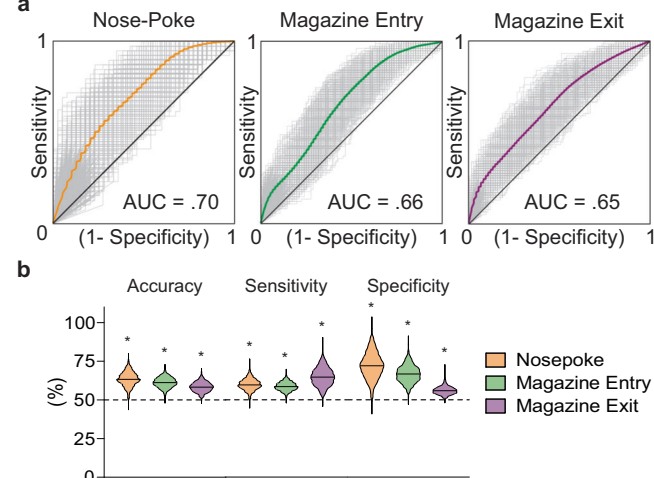

**Fig. 3 | SVM modeling predicts goal-directed and habitual reward seeking using external globus pallidus (GPe) arkypallidal neuron Ca²⁺ dynamics. a** Averaged receiver operator characteristic (ROC) curves for prediction of goal-directed or habitual reward-seeking around behavioral events (nose-poke, magazine entry, and magazine exit; −2 – 2 s input range). **b** Comparison of accuracy, sensitivity, and specificity to random chance among each behavioral event. Two-tailed one sample $t$ tests were used for (**b**). For accuracy NP versus 50%, $t = 116.1$, $p < 0.0001$, for accuracy M$_{entry}$ versus 50%, $t = 136.4$, $p < 0.0001$, for accuracy M$_{exit}$ versus 50%, $t = 105.7$, $p < 0.0001$. For sensitivity NP versus 50%, $t = 109.1$, $p < 0.0001$, for sen-sitivity M$_{entry}$ versus 50%, $t = 126.9$, $p < 0.0001$, for sensitivity M$_{exit}$ versus 50%, $t = 101.4$, $p < 0.0001$. For specificity NP versus 50%, $t = 104.8$, $p < 0.0001$, for spe-cificity M$_{entry}$ versus 50%, $t = 118.9$, $p < 0.0001$, for specificity M$_{exit}$ versus 50%, $t = 94.35$, $p < 0.0001$. *$p < 0.05$ for comparison to random chance for accuracy, sensitivity, and specificity. See Supplementary Table 1 for AUC values for additional time input ranges. See Supplementary Table 2 for full statistical information. Source data are provided as a Source Data file.

For magazine exit, GPe arky Ca²⁺ signal is decreased for the rewarded events in RR and for the unrewarded events in RI (linear regression, $p < 0.05$; Supplementary Fig. 3e, f).

## GPe arkypallidal neuronal activities have information sufficient to distinguish goal-directed and habitual-seeking behavior

To further examine if the neural activity contains reward-seeking strategy information, we trained a support vector machine (SVM), a widely used supervised machine learning algorithm. SVMs produce robust predictions with minimal risk of overfitting and has demon-strated utility in analyzing neural activity data[28–31]. All trials from both RR and RI were pooled together, and SVM was trained to classify the neural data as RR or RI. The more GPe arky Ca²⁺ signal encodes

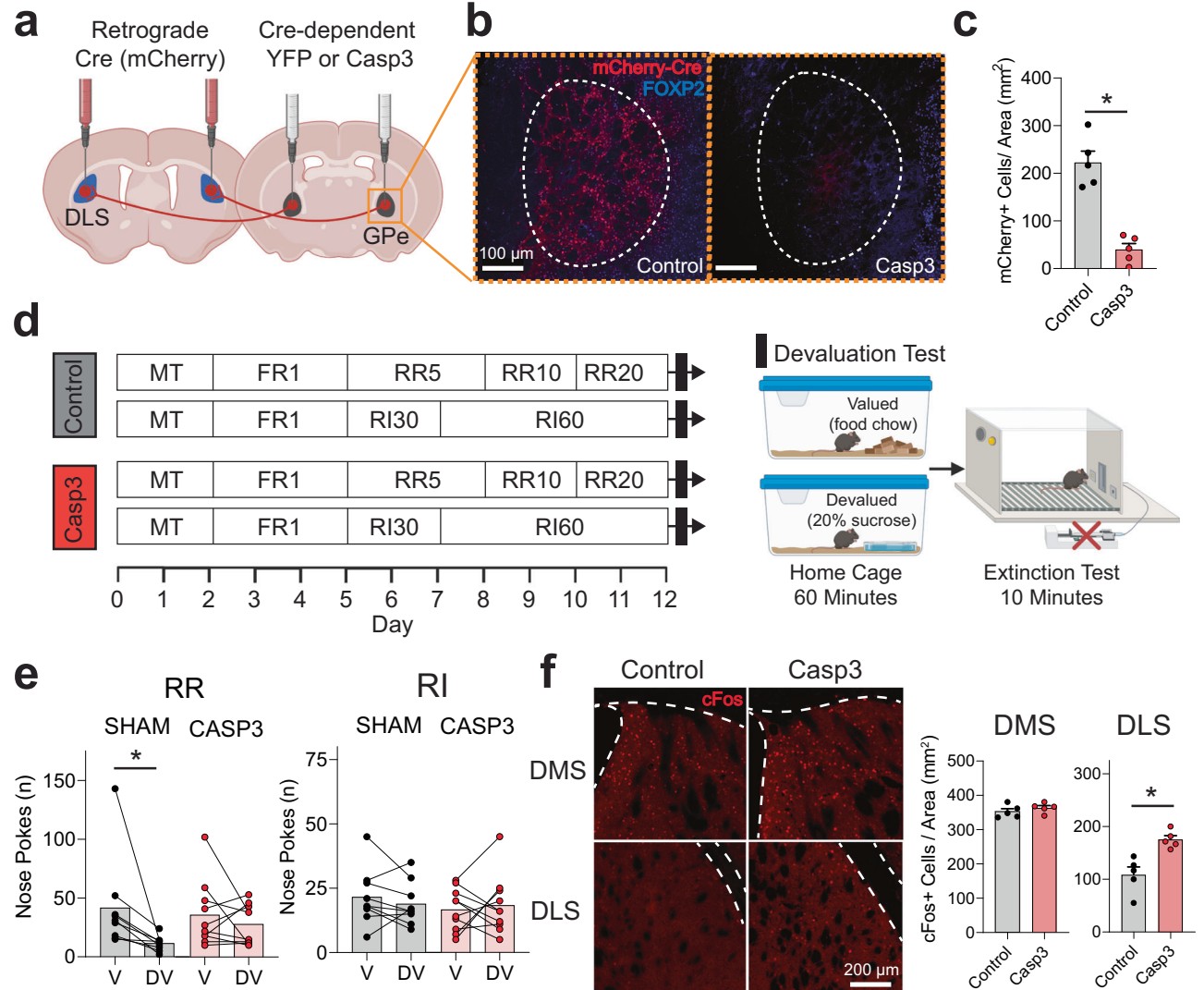

**Fig. 4 | Effects of Caspase 3 (casp3)-dependent ablation of external globus pallidus (GPe) arkypallidal neurons on goal-directed and habitual-seeking.** **a** Schematic of virus injections for Cre-dependent ablation of GPe arkypallidal neurons. **b** Representative GPe images of sham control and caspase mice (Scale: 100 μm) from *n* = 5 mice/group. **c** Caspase mice showed a significant reduction of mCherry-positive neurons in the GPe (*t* = 6.90, *p* = 0.001). *n* = 5/group. **d** Operant RR and RI behavior schedules in sham control and caspase mice to establish goal-directed and habitual seeking, respectively. **e** RR-trained (Goal-directed) sham mice reduced nose poke responses in the devalued state, confirming goal-directed behavior (*p* = 0.0039). However, caspase mice showed no changes in nose poke responses between valued and devalued states, typical of habitual behavior

(*p* = 0.73). No differences in nose poke rates between the valued and devalued state for RI-trained sham (*p* = 0.38) and caspase mice (*p* = 0.83) indicated habitual reward-seeking. *n* = 10 mice/group. **f** Representative IHC images of cFos expression in the dorsomedial (DMS) and dorsolateral (DLS) striatum for sham and casp3 mice (Scale: 200 μm). GPe arkypallidal neuron ablation increased the number of cFos-positive cells in the DLS (*t* = 3.99, *p* = 0.004), but not DMS (*t* = 1.02, *p* = 0.34). *n* = 5 mice/group. Two-tailed *t* tests were used for (**c**, **f**). Wilcoxon test was used for (**e**). Data represent mean ± SEM. *\**p* < 0.05. See Supplementary Table 2 for full statistical information. Source data are provided as a Source Data file. (**a**, **d**) were created with BioRender.com.

reward-seeking strategy, the better the SVM predictions. To accommodate the temporal dynamics of GPe arky neurons, we utilized neural activity 2 s before and after the behavioral events. Prediction accuracy of SVM models was estimated with the area under the curve (AUC) of their receiver operating characteristic (ROC) curves, accuracy, specificity, and sensitivity. Our SVM showed that GPe arky Ca²⁺ signal around all three behavioral events contained enough information to distinguish RR vs RI reward-seeking strategy (Fig. 3a). Using different Ca²⁺ input time ranges associated with behaviors, we confirmed the validity of SVM classification (Supplementary Table 1). Accuracy, sensitivity, and specificity of our SVM were also significantly greater than chance (one-sample *t* test, *p* < 0.05; Fig. 3b). Although we have not measured GPe arky Ca²⁺ signal during devaluation, given that both neural activity and reward-seeking behavior

gradually develop through training, our results suggest that GPe arky Ca²⁺ signal conveys action-selection information underlying goal-directed and habitual behavior.

## Caspase3-dependent ablation of GPe arkypallidal projections to the DLS shift mice towards habitual behavior

To determine whether ablation of this arky^GPe→DLS circuit modulates goal-directed or habitual behavior, we used a Cre-dependent caspase 3 virus which induces cell-autonomous death with minimal toxicity to neighboring cells[32–36]. We bilaterally injected an mCherry-tagged retrograde virus expressing Cre recombinase [AAV-Ef1a-mCherry-IRES-Cre] into the DLS, followed by a second injection of Cre-dependent caspase-3 (AAV5-flex-taCasp3-TEVp; or control AAV5-Ef1a-DIO EYFP) into the GPe (Fig. 4a). We validated a significant reduction in mCherry-positive

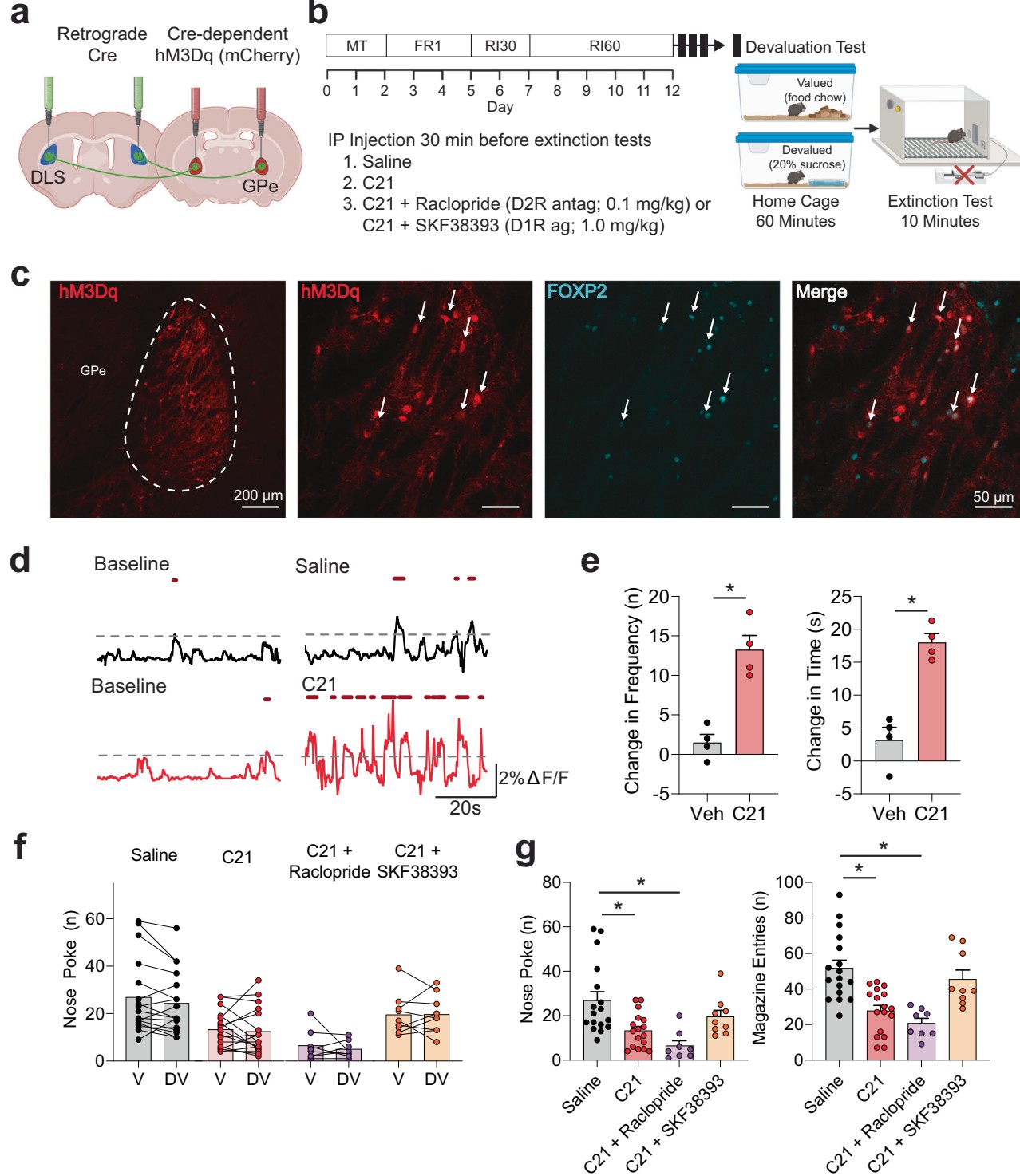

**Fig. 5 | Effects of chemogenetic activation of external globus pallidus (GPe) arkypallidal neurons on habitual reward-seeking. a** Schematic of virus injection strategy for Cre-dependent expression of the excitatory DREADDs hM3Dq. **b** RI operant schedule followed by evaluation tests with IP injection groups. **c** Representative images showing hM3Dq expression in the GPe and coexpression with arkypallidal neuron marker FOXP2 from $n = 3$ mice. Scale: 200 μm (left), 50 μm (rest). **d** Representative calcium traces for hM3Dq-expressing GPe arkypallidal neurons at baseline and following C21 or vehicle administration in freely moving mice. **e** C21 administration caused a notable increase in the frequency and duration of $Ca^{2+}$ events (frequency: $t = 4.47$, $p = 0.021$, time: $t = 11.86$, $p = 0.0013$). $n = 4$ mice/group. **f** There were no differences between the valued and devalued states for IP injection treatment. **g** The effects of DREADDs activation and IP injection groups on

nose poke and magazine-entry seeking behaviors during the extinction test. Noke poke with drug treatment ($p = 0.0004$), $z = 2.71$, $p = 0.04$ for saline versus C21, $z = 3.96$, $p = 0.0005$ for saline versus C21+Raclopride, $z = 0.83$, $p = 0.99$ for saline versus C21 + SKF383393. Magazine entry with drug treatment ($p < 0.00001$), $z = 3.59$, $p = 0.002$ for saline versus C21, $z = 4.13$, $p = 0.0002$ for saline versus C21+Raclopride, $z = 0.69$, $p = 0.99$ for saline versus C21 + SKF383393. Two-tailed $t$ tests were used for (**e**). Wilcoxon test was used for (f). Kruskal–Wallis test with Dunn's posthoc tests were used for (**g**). $n = 17$ (Saline, C21), 8 (C21 + Raclopride), 9 (C21 + SKF38393) for (**f, g**). Data represent mean ± SEM. *$p < 0.05$. See Supplementary Table 2 for full statistical information. Source data are provided as a Source Data file. (**a, b**) were created with BioRender.com.

neurons in the GPe of the caspase group (unpaired *t* test, *p* < 0.05; Fig. 4b, c).

Since previous studies have implicated GPe arky neurons being associated with motor function[20,21], we examined if GPe arky neuron ablation resulted in motor dysfunction. In the open field test, we observed no significant differences in spontaneous locomotion, velocity, ambulatory time, or ambulatory episodes, indicating that partial arky$^{GPe→ DLS}$ circuit ablation does not alter basic motor function (Mann–Whitney test, *p* > 0.05; Supplementary Fig. 4a). In the first 10 min, we found no significant changes in time in the center or periphery zone, or entries into center or periphery zone (*p* > 0.05; Supplementary Fig. 4b), suggesting no observable impact on anxiety-like behavior. To assess dorsal striatum-dependent motor learning, we utilized an accelerated rotarod paradigm which has previously been shown to result in DLS-dependent skill acquisition[37]. Both groups learned the task well, indicated by a significant increase in latency to fall across training sessions (Two-way RM ANOVA, *p* < 0.05; Supplementary Fig. 4c) without significant overall group differences in latency to fall, nor were there any group differences in the change across sessions (*p* > 0.05; Supplementary Fig. 4c). However, for daily average latency to fall values, we found an interaction between group differences and the day of testing (*p* < 0.05; Supplementary Fig. 4d). The caspase group had a shorter latency to fall time for days 1 and 2 (*p < 0.05*; Supplementary Fig. 4d), but similar in the remaining three training days (days 3–5), indicating that partial arky$^{GPe→ DLS}$ circuit ablation may slow the initial motor learning without long-term effects.

During magazine training, both control and caspase mice showed reduced latency to the magazine from the first day to the last for both RR and RI (Two-way RM ANOVA, *p* < 0.05; Supplementary Fig. 5b) without differences between caspase and control mice for RR or RI groups. During operant training, mice showed increased nose poke rates for both RR and RI (*p* < 0.05; Supplementary Fig. 5c) schedules in both the caspase and control groups. Altogether our results demonstrate arky$^{GPe→ DLS}$ circuit ablation does not impair overall performance or acquisition rate in the operant reward-seeking task. In the devaluation test, control mice in the RR group showed a reduction in extinction session nose pokes for the devalued state (Wilcoxon test, *p* < 0.05; Fig. 4e), consistent with goal-directed behavior. Interestingly, RR caspase mice exhibit habitual behavior (*p* > 0.05; Fig. 4e), indicating that loss of arky$^{GPe→ DLS}$ circuit function promotes a shift from goal-directed to habitual behavior. In contrast, RI caspase mice showed no significant differences between the valued and devalued states in both sham control and caspase mice (*p* > 0.05; Fig. 4e). Supporting that GPe arky ablation disinhibits DLS cellular activities, caspase mice had a significantly higher number of cFos-positive cells in the DLS compared to the control group, but not in the DMS (*p* < 0.05; Fig. 2f).

To determine whether this shift towards habitual behavior is specific to sucrose reward-seeking, we trained an additional set of mice with a 10% sucrose and 10% ethanol solution reward (10S10E). Similar to the sucrose reward-seeking paradigm, in the devaluation extinction test, we found no difference between the valued and devalued states for caspase mice on the RR operant schedule (*p* > 0.05; Supplementary Fig. 6a). Also, both RI caspase and sham control mice exhibited no differences between the valued and devalued states (*p* > 0.05; Supplementary Fig. 6b). To determine whether this shift to habitual behavior is possibly due to a change in motivation or valuation of the reward, or a more specific reinforcement learning process, we compared 10% ethanol preference and consumption between control and caspase mice in a 24 h continuous-access two-bottle choice paradigm. We found no difference in 10% ethanol preference or consumption (Two-way RM ANOVA, p > 0.05; Supplementary Fig. 6c) between the caspase and control mice, suggesting that habitual seeking behavior is not necessarily correlated to reward preference.

## Chemogenetic activation of GPe arkypallidal neurons reduces overall seeking-behaviors during devaluation extinction testing

To determine if activation of GPe arky neurons could inhibit or reverse RI habitual behavior, we selectively expressed the Gq-coupled designer receptors exclusively activated by designer drugs (DREADDs) in arky neurons by first injecting a retrograde virus expressing Cre recombinase into the DLS (AAV.hSyn.HI.eGFP-Cre.WPRE.SV40). Next, we injected a Cre-dependent hM3Dq DREADDs virus into the GPe [AAV5-hSyn-DIO-HM3D(Gq)-mCherry; Fig. 5a] and trained mice on an RI schedule (Fig. 5b). We confirmed DREADDs expression in arky neurons via the overlapping of the mCherry with the FOXP2 cellular marker (Fig. 5c). We next examined the dose-dependent effects of C21 (veh, 0.5, 1.0, 2.0 mg/kg) on locomotor function in the open field test. We found no effects of C21 concentration on locomotor or anxiety-like behaviors in the open field test (Kruskal–Wallis test, *p* > 0.05; Supplementary Fig. 7a, b). Based on this and previous work in our lab, we selected 1.0 mg/kg injections of C21 *i.p.* and confirmed an increase in Ca$^{2+}$ signal events and duration above a 2% threshold in freely moving mice (Fig. 5d, e)[38]. As arky neurons have been shown to inhibit both dopamine 1 receptor (D1R) and dopamine 2 receptor (D2R)-expressing neurons in the dorsal striatum[17,19–21], we sought to determine whether any behavioral changes primarily occurred via D1R- or D2R- dependent mechanisms by testing combined C21 + Raclopride (D2R antagonist; 0.1 mg/kg) and C21 + SKF38393 (D1R agonist; 1.0 mg/kg) injection groups. Again, we tested these injection groups in the open field test. We found a main effect of injection group on ambulatory episodes (Kruskal–Wallis test, *p* < 0.05; Supplementary Fig. 8a), but did not find any group differences compared to the control injections (posthoc Dunn's test, *p* > 0.05; Supplementary Fig. 8a). We did not find any other differences in locomotor or anxiety-like behaviors (Kruskal–Wallis test, *p* > 0.05; Supplementary Fig. 8a, b). During magazine training, mice showed reduced latency to the magazine from the first day to the last and increased nose poke rates across training sessions (Two-way RM ANOVA, *p* < 0.05; Supplementary Fig. 9b, c). To test the effect of arky activation on habitual behavior, C21 (1 mg/kg *i.p.*) was administered 30 min before the extinction test for the valued and devalued states. In the devaluation test, we observed no significant differences between the valued and devalued states for saline, C21, C21 + raclopride, nor C21 + SKF (Wilcoxon test, *p* > 0.05) injection groups (Fig. 5f). However, the C21 injection significantly reduced nose pokes and magazine entries (Dunn's posthoc test, *p* < 0.05; Fig. 5g) compared to the saline injection. Coadministration of C21 and raclopride reduced nose poke, magazine entries, and magazine duration (*p* < 0.05; Fig. 5g) compared to saline injections. Interestingly, only coadministration of the D1R agonist, SKF38393, prevented C21-induced reductions in seeking behaviors (*p* > 0.05; Fig. 5g), indicating behavioral effects of arky activation may primarily be through D1R-expressing dMSNs.

Given that SKF38393 prevented C21-induced reductions in seeking behaviors, we next examined the effect of SKF38393 alone on habitual behaviors. In another cohort, systemic D1R agonism did not affect habitual responding in the devaluation task (Wilcoxon test, *p* > 0.05; Supplementary Fig. 10a). SKF38393 did significantly increase magazine entries during the extinction session compared to saline and C21 + SKF groups (Dunn's posthoc test, *p* < 0.05; Supplementary Fig. 10b). However, there were no differences in nose pokes or magazine duration (Kruskal–Wallis test, *p* > 0.05; Supplementary Fig. 10b). Given that C21 caused global reductions in seeking behavior and did not affect reward devaluation, we also tested the effects of arky activation on goal-directed behavior. Saline and C21 injections of RR-trained mice both showed significant decreases in nose pokes in the devalued state, indicating goal-directed behavior (Wilcoxon test, *p < 0.05*; Supplementary Fig. 11a). Similar to chemogenetic activation in RI habitual mice, C21 activation of GPe arky neurons significantly reduced nose poke, and magazine entry behaviors during the extinction test (Mann–Whitney test, *p* < 0.05; Supplementary Fig. 11b), again

indicating that arky activation may lead to a global reduction in seeking-behaviours. Lastly, we again sought to determine if these effects were reward-specific and trained mice on an RI schedule with a 10S10E reward. Again, C21 had no effect on reward devaluation (Wilcoxon test, $p > 0.05$; Supplementary Fig. 12a), but did significantly reduce seeking behaviors during the extinction test (Mann–Whitney test, $p < 0.05$; Supplementary Fig. 12b).

## Discussion

In the present study, we provide a role in the function of the arky$^{GPe \to DLS}$ circuit showing that GPe arky neurons exhibit notably distinct activity patterns between RR and RI operant schedules, which develop goal-directed and habitual behavior, respectively. In addition, targeted genetic ablation of arky$^{GPe \to DLS}$ circuit promoted a transition from goal-directed reward to habitual seeking behaviors. In contrast, chemogenetic activation led to a global reduction in seeking behaviors, which was nullified by D1R agonism as we illustrated in Supplementary Fig. 13.

Our findings initially seemed to contradict since we observed an increased $Ca^{2+}$ signal during the habit training. However, it is important to note that the behavioral shaping time points in RI120 and RR20 are different than the actual assessment of goal-directed and habitual behaviors through devaluation and extinction. One possibility is that GPe arky activities during the behavioral training represent a counteracting signal to DLS activities that are more prominent in habitual behavior[17,20,21,39]. Consequently, higher GPe arky activities during RI habit shaping may also underlie the optimization of action selection in the distinct random operant schedules. RR operant conditioning maintains the action-outcome association and administers reward directly proportional to the animal's effort and response rates. However, RI conditioning is solely time-interval based and does not deliver increased reward with higher nose-poke rates[11,13,24,40–42]. Thus, the optimal effort for reward during RI conditioning is for animals to suppress seeking behaviors between the random time intervals. This is further supported by reduced nose-poke rates in the RI operant schedules compared to RR (Supplementary Figs. 2c, 5c), as well as the suppressed seeking behaviors following chemogenetic activation (Fig. 5g, Supplementary Figs. 10b, 11b, 12b). On the other hand, it is also possible that GPe arky neurons could support behavioral flexibility and action selection through an inverted U-curve relationship frequently seen in cortical dopamine signaling and cognitive control in other studies[43–45]. Future work examining simultaneous activities in GPe arky and DLS medium spiny neurons (MSNs) may clarify the competing striatal "go" and arky "stop" relationship in habitual behavior.

Arky neurons suppress dorsal striatum-dependent behaviors through GABA release onto striatal neurons[20,21]. Indeed, our study demonstrated that arky$^{GPe \to DLS}$ circuit ablation increased cellular activity markers in the DLS, but not the DMS (Fig. 4f). Similarly, previous whole-GPe lesions led to increased dopamine-induced striatal cFos expression[46]. Circuit studies have shown that arky neurons project to fast-spiking interneurons as well as both direct and indirect MSNs (dMSNs, iMSNs) in the dorsal striatum[6,17,19]. We showed that chemogenetic activation of GPe arky neurons overall reduced seeking behaviors, but it remained unknown which cellular target may primarily mediate these behavioral effects. Notably, only systemic coadministration of C21 and the D1R agonist SKF38393 prevented C21-induced reductions in habitual seeking-behaviours (Fig. 5g). This suggests that arky neurons may suppress seeking behaviors primarily through D1R-expressing dMSNs in the DLS. This is consistent with previous findings showing that habit development is accompanied by strengthening of both dMSNs and iMSNs in the DLS, whereas habit suppression was solely associated with weakened dMSNs output[39]. Our findings are limited due to the systemic injection of the D1R agonist. Future studies with pharmacologic microinjections or optogenetic manipulations of DLS dMSNs and iMSNs will validate the precise impact of arky neurons in the DLS. Ex vivo electrophysiology studies will also be beneficial to

characterize these circuit-specific effects and determine if GPe arky activities can alter neuroplastic changes in the DLS including potentiation of corticostriatal glutamatergic synapses that accompany habit development[11,22,24,40,47]. The current study targeted arky neurons via a projection-dependent strategy using retrograde viruses injected into the DLS and showed significant overlap with the FOXP2 cellular marker. While all GPe cells that express FOXP2 project back to the striatum, small numbers of non-FOXP2+ pallidostriatal neurons have been identified, which include NPAS1+, LHX6+, or even a small percentage of PV+ neurons[15,16]. It is possible that these sub-populations of arky neurons may have distinct dorsal striatum targets. Thus, future studies are warranted to complement these projection-based studies using various Cre- mouse lines or promoter-driven viruses to determine if arky subtypes preferentially interact with dMSNs, iMSNs, or other interneurons in the dorsal striatum.

While we provide evidence that the arky$^{GPe \to DLS}$ circuit regulates goal-directed and habitual behavior, it remains unclear whether reward prediction and action-outcome contingencies are computed in the GPe or in its afferent projections. The largest innervation of the GPe is from GABAergic iMSNs in the dorsal striatum and glutamatergic neurons in the STN[1,11,13]. While the DMS and DLS have not been documented to have direct communication, our studies provide a possibility that arky neurons could form a more direct circuit between dorsal striatal regions and directly regulate the transition between goal-directed and habitual behavior. Dorsal striatum iMSNs have been shown to affect GPe arky activities directly and indirectly through a di-synaptic circuit involving GPe prototypic neurons[20,48]. However, it has not been studied in the context of dorsal striatum lateralization nor for habitual behavior. A small portion of dMSNs also reportedly project to the GPe as well and could contribute to arky neuron activities[48]. Interestingly, M1 and M2 motor cortex neurons have been shown to contain glutamatergic projections onto GPe arky neurons, but its functional effects on behavior have not been investigated[49]. As the DLS also receives significant glutamatergic innervation from the motor cortex, we may need to address if there is an overlap or separation of motor cortex neurons that directly project to the DLS or GPe arky neurons. Modified rabies virus trans-synaptic tracing could prove particularly useful in characterizing di-synaptic circuits through GPe arky neurons, as well as cell-specific tracing in Npas1- or Foxp2-Cre mice[50–52].

Increasing evidence indicates that GPe plays a critical role in non-motor functions, including action selection, reward learning and prediction, and behavioral flexibility[1,6]. We showed that GPe arky neurons preferentially target the DLS compared to the DMS, which is known to regulate habitual reward-seeking and motor-skill acquisition[13,22,39,53,54]. Previous behavioral studies investigating GPe arky neurons have focused on motor function and show an important role for behavioral and locomotive suppression[20,21]. Thus, when examining the nonmotor function of GPe arky neurons, it is essential to document any effects manipulations may have on basic motor function which could confound results. We show that arky$^{GPe \to DLS}$ genetic ablation did not affect locomotor or anxiety-like behaviors in the open field test (Supplementary Fig. 4a, b). We did observe some minor deficits during the first two days of accelerated rotarod; however, caspase animals recovered to control levels by day three and beyond (Supplementary Fig. 4d). Despite this, we importantly did not find any deficits during operant conditioning acquisition (Supplementary Fig. 5b, c).

We primarily utilized sucrose for our behavioral studies as a reward. While it is not completely representative of addictive drugs, sucrose is a highly motivating reward for operant conditioning in rodents and can produce many of the characteristics of substance use disorders, including inflexible reward-seeking and an over-reliance on maladaptive habitual behaviors[23,24,26,40,55,56]. We showed that loss of GPe arky neurons resulted in inflexible habitual seeking and insensitivity to reward devaluation. This suggests that reduced or dysfunctional arky

activities could contribute to substance use disorders in humans and that their activation or restoration of function could be a potential therapeutic strategy for addiction. Like the sucrose experiments, we found the same shift towards habitual behavior with arky deletion and reduction in habitual seeking behaviors following chemogenetic activation using an ethanol-containing reward (Supplementary Figs. 6a, 12b). This supports the generalizability of our findings for more addictive rewards and specifically alcohol use disorder. Investigating sex-specific functions of GPe arky neurons will also be critical to this generalizability given documented sex differences in habit formation and addictive behaviors[57–59]. Interestingly, we did not find any baseline differences in ethanol preference or consumption in a two-bottle choice paradigm following GPe arky cell deletion (Supplementary Fig. 6c). This indicates that arky neurons may specifically regulate habitual seeking and habit development as opposed to the valuation of the reward itself. Additional studies may reveal the effects of chronic stress or ethanol exposure on arky regulation of the DLS. Chronic ethanol exposure has been shown to disrupt top-down regulation of the dorsal striatum by the OFC and produce habitual seeking and could similarly explain how dysfunctional bottom-up regulation by arky neurons could lead to maladaptive habitual behaviors[60].

Overall, our study found that GPe arky neurons play a critical role regulating goal-directed and habitual-seeking behaviors. Not only did we show distinct activity patterns in GPe arky activity during goal/habit shaping, but targeted manipulation of the arky$^{GPe \to DLS}$ circuit revealed an important role in suppressing reward-seeking. Further, diminished arky$^{GPe \to DLS}$ function disinhibits DLS activity and leads to an inability to properly inhibit reward-seeking upon devaluation. These findings may represent a promising therapeutic strategy for treating addiction and other compulsive disorders.

## Methods

### Animals
All experimental procedures were approved by the Mayo Clinic Institutional Animal Care and Use Committee and performed following NIH guidelines. C57BL/6J mice were purchased from Jackson Laboratory (Bar Harbor, ME). Mice were housed in standard Plexiglas cages. The colony room was maintained at a constant temperature ($24 \pm 1\,^\circ C$) and humidity ($60 \pm 2\%$) with a 12 h light/dark cycle (lights on at 07:00 A.M.). We used 8- to 10-week-old male mice for all experiments. Mice were allowed *ad libitum* access to food and water. For the operant conditioning tests, mice were food restricted to 85% of their baseline weight, at which time they were maintained for the duration of experimental procedures.

### Operant conditioning
We conducted operant conditioning using the same operant chambers/schedules as our previous studies[38,61,62]. Briefly, mice were placed in operant chambers (Med-Associates, St Albans, VT) in which they poke a single hole for an outcome of 20% sucrose (dissolved in tap water; 10 μl per reinforcement). For some experiments, sweetened ethanol (10% sucrose, 10% ethanol) was used as the reward. Before training, mice were food restricted to approximately 85% body weight, which was maintained for the duration of experimental procedures. Food restriction body weights are presented in Supplementary Figs. 2, 5, 9 On the first day, mice were trained to approach the reward magazine with a reward delivered on a random time schedule for 30 min. Next, mice were trained on a fixed ratio 1 schedule for 1 h in 3 sessions. After acquiring nose-poking behavior, mice were trained on random interval (RI30 2 days/RI60 2–3 days/RI120 4 days) to develop habitual behavior or random ratio (RR2 1–2 days/RR5 2–3 days/RR10 2–3 days/RR20 2–3 days) to form goal-directed reward-seeking. Sessions were completed after 30 min or following 60 reward reinforcements. RI30/60/120 delivered one reward outcome on average every 30/60/120 s after the last reward outcome to develop habitual reward-seeking. RR2/5/10/20 delivered one reward outcome on average every 2/5/10/20 response in the correct nose-poke hole to develop goal-directed reward-seeking.

### Reward devaluation and extinction test
On the devalued day, mice were given 1 h of *ad libitum* access to the outcome (20% sucrose or 10% ethanol/10% sucrose) previously earned by nose poking (for devaluation) or food pellets, and then underwent serial nonreinforced extinction sessions in each training context. The order of the valuation context was counterbalanced across mice and was 10 min in duration. Drug treatments were administered intraperitoneally (*i.p.*) 30 min prior to each extinction session.

### Stereotaxic surgery for virus injection
Mice were anesthetized with isoflurane (1.5% in oxygen gas) using a VetFlo™ vaporizer with a single-channel anesthesia stand (Kent Scientific Corporation, Torrington, CT) and placed on the digital stereotaxic alignment system (model 1900; David Kopf instruments, Tujunga, CA). Hair was trimmed, and the skull was exposed using 8-gauge electrosurgical skin cutter (KLS martin, Jacksonville, FL). The skull was leveled using a dual-tilt measurement tool. Holes were drilled in the skull at the appropriate stereotaxic coordinates. Viruses were infused to the DLS (AP + 0.7 mm, ML ± 2.5 mm, DV −3.1 mm from bregma) or GPe (AP −0.4 mm, ML ± 2.0 mm, DV −3.8 mm from bregma) at 100 nl/min for 4 min (400 nl total) through a 33-gauge injection needle (cat # NF33BV; World Precision Instruments, Sarasota, FL) using a microsyringe pump (Model UMP3; World Precision Instruments). The injection needle remained in place for an additional 5 min following the end of the injection. All viruses were purchased from Addgene (Watertown, MA), and were injected at the following titers: AAV5-CaMKII(1.3).eYFP.WPRE.hGH ($1 \times 10^{13}$ vg/mL), retrograde AAV-hSyn1-GCaMP6s-P2A-nls-dTomato ($7 \times 10^{12}$ vg/mL), retrograde AAV-Ef1a-mCherry-IRES-Cre ($7 \times 10^{12}$ vg/mL), retrograde AAV.hSyn.HI.eGFP-Cre.WPRE.SV40 ($7 \times 10^{12}$ vg/mL), AAV5-flex-taCasp3-TEVp ($7 \times 10^{12}$ vg/mL), AAV5-Ef1a-DIO EYFP ($1 \times 10^{13}$ vg/mL), AAV5-hSyn-DIO-hM3D(Gq)-mCherry ($7 \times 10^{12}$ vg/mL). Following stereotaxic surgery, we injected buprenorphine sustained release (1 mg/kg, *s.c.*; ZooPharm, Laramie, WY) to alleviate post-surgery pain.

### Immunofluorescence
Brains were fixed with 4% paraformaldehyde (Sigma-Aldrich, St. Louis, MO) and transferred to 30% sucrose (Sigma-Aldrich) in phosphate-buffered saline at 4 °C for 72 hr. Brains were then frozen in dry ice and sectioned at 40 μm using a microtome (Leica Corp., Bannockburn, IL). Brain slices were stored at −20 °C in a cryoprotectant solution containing 30% sucrose (Sigma-Aldrich) and 30% ethylene glycol (Sigma-Aldrich) in phosphate-buffered saline. Sections were incubated in 0.2% Triton X-100 (Sigma-Aldrich), 5% bovine serum albumin in phosphate-buffered saline for 1 hr, followed by incubation with the primary antibody in 5% bovine serum albumin overnight at 4 °C. After three washes in phosphate-buffered saline, the sections were mounted onto a glass slide coated with gelatin and cover-slipped with a VECTASHIELD® antifade mounting medium (Vector Laboratories, Burlingame, CA). Images were obtained using an LSM 780 laser scanning confocal microscope (Carl Zeiss, Heidelberg, Germany) using a 10x or 40x water-immersion lens. Relative fluorescence values were determined using Image J (1.53t, National Institute of Health, USA). We used 488 (eGFP), 568 (mCherry), and 405 channels for imaging. All antibodies were purchased from Abcam (Cambridge, UK), and included anti-FOXP2 (rabbit; ab16046; 1:500) and anti-parvalbumin (rabbit; ab11427; 1:500) for primary, and anti-rabbit Alexa Fluor® 405 (donkey; ab175651; 1:500).

### In vivo Ca$^{2+}$ signal with fiber-photometry
We recorded the cellular Ca$^{2+}$ transients in real-time in vivo using single-channel fiber-photometry with a CineLyzer system (Ver. 1.0.1,

Plexon, Dallas, TX)[38,63]. We implanted a fiber-optic cannula (200/240 μm diameter, 200 μm end fiber) into the GPe (AP−0.46 mm, ML +2.0 mm, DV −3.8 mm from bregma) of mice injected with the AAV retrogradely expressing GCaMP6s in the DLS. The implanted fiber was linked to a patch cord, and the light intensity at the fiber tip was 60 μW consistently. These output signals were projected onto a CCD camera-type photodetector by the same optical fiber, passed through a GFP filter, and collected at 30 frames per second. For analysis of the photometry data, in the CineLyzer system, the calculation of $\Delta F/F$ values was conducted through three different stages[63,64] and exported to MATLAB. First, the raw fluorescence $F_{RAW}(t)$ was averaged to get $F_{AVG}(t)$ over a sliding time window (0.75 s). The baseline for a particular frame $F_{BASELINE}(t)$ was the minimum of the $F_{AVG}(t)$ in a time window (3 s) preceding this frame. Second, $\Delta F/F(t)$ was calculated by subtracting $F_{BASELINE}(t)$ from $F_{RAW}(t)$ and then dividing it by $F_{BASELINE}(t)$. Finally, $\Delta F/F(t)$ was smoothed with an exponentially weighted moving window, which is described by a time constant, $\tau$ (0.2 s), and a width, $w$ (1 s). For operant conditioning, fiber photometry was performed during the last sessions of RR20 and RI120. For chemogenetic validation, the $Ca^{2+}$ signal was recorded for 10 min following systemic saline or C21 injection. Fiber photometry data were completed with RI animals first ($n = 5$) followed by RR ($n = 5$).

## Classification analyses

To estimate the amount of task-related information in GPe arky $Ca^{2+}$ signal, we used a support vector machine (SVM). SVM is a supervised machine learning algorithm for classification and regression analyses. SVMs have been widely used because they produce robust predictions with minimal risk of overfitting and have demonstrated utility in analyzing neural activity data[28–31]. We trained an SVM to predict reward-seeking strategy (RR or RI) using average arky $Ca^{2+}$ signal during the 2 s period prior to and following each behavioral event (NP, magazine entry, magazine exit). We trained the SVM with 4-fold cross-validation. To balance the dataset, we performed random under sampling. Random under sampling was repeated 400 times for each analysis and an averaged prediction performance of the SVM was calculated. To measure the prediction accuracy of the SVMs, we used area under the curve (AUC) of the SVM's receiver operator characteristic (ROC) curves. As the prediction accuracy of the SVM increases, the AUC gets closer to 1 (AUC of 0.5 = random chance; AUC of 1.0 = 100% accuracy). To measure the SVMs' prediction accuracy, we also use accuracy [(tn + tp)/(tn+ tp + fn + fp)], sensitivity [tp/(tp + fn)], and specificity [tn/(tn+fp)] which indicate how well the model predicts, detects true positives (tp), and detects true negatives (tn), respectively.

## Caspase 3-mediated circuit ablation

To ablate the circuit in a targeted fashion, we used a genetically engineered caspase 3, whose activation commits a cell to apoptosis[34]. Endogenous caspase 3 exists as procaspase 3, which is cleaved into its active form by upstream apoptotic signals and other caspase proteins. This genetically engineered caspase lacks the cleavage site for upstream caspases and can only be cleaved by a tobacco etch virus protease (TEVp) which is coexpressed in an AAV[33]. AAV-flex-taCasp3-TEVp was expressed in a cre-dependent manner in the GPe to only ablate cells that express Cre recombinase from retrograde injections in the DLS (AAV-Ef1a-mCherry-IRES-Cre). Importantly, caspase 3 triggers cell-autonomous apoptosis, minimizing the risk of off-target effects and toxicity to neighboring cells[32,35,36]. Caspase 20% sucrose operant conditioning experiments were performed with RI control/casp3 first ($n = 10$/group), followed by RR control/casp3 ($n = 10$/group). The 10S10E reward replicate was completed in a second cohort ($n = 8$/group).

## Chemogenetics and drug treatments

We purchased compound 21 from Hello Bio (C21; Princeton, NJ). Based on previous experiments[38], we administered C21(1 mg/kg), dopamine

receptor 1 agonist SKF38393 (1.0 mg/kg), dopamine receptor 2 antagonist raclopride (0.1 mg/kg), or saline *i.p.* 30 min before the experiments in mice. These concentrations have been previously shown to alter reward-seeking behaviors with minimal motor effects or risk of seizure[43,65–69]. Chemogenetic 20% sucrose operant conditioning experiments were first completed in RI (2 cohorts, $n = 9$, then $n = 8$). Then the 10S10E replicate was completed in a second cohort ($n = 10$). Lastly, RI SKF ($n = 8$) and goal-directed experiments were completed ($n = 8$).

## Open field test

The open-field test (OFT) was conducted in chambers (Med-Associates, St Albans, VT) to measure the locomotor responses of mice. The session lasted 30 min and total distance and velocities were recorded using beam breaks. The first 10-min bin was referenced to measure anxiety-like behavior. Time spent in the open zone (%) was measured as time spent in the open zone / total time x 100. Animals were habituated to the testing room in their home cage 1 h prior to behavioral testing.

## Rotarod

A computer-interfaced rotarod accelerating from 4–40 rotations per min over 300 s was used. Animals were trained with ten trials per day for 5 days (trained every other day). This training protocol was based on previous behavioral, pharmacological, and electrophysiological studies that showed this extended training resulted in DLS-dependent skill acquisition[37]. Animals were habituated to the testing room in their home cage 1 h prior to behavioral testing.

## Two-bottle choice

Oral ethanol consumption and preference were examined using a two-bottle choice test in the mouse home cage. Mice were individually housed and given 24-h access to two bottles: water and ethanol. The concentration of ethanol was raised from 3 to 6% to 10% ethanol (10E, v/v) on every 4th day to adapt ethanol intake. Every other day the bottle placement was switched to prevent place preference development. After increasing the ethanol concentration to 10% the mice had 14 days of 10E access. Ethanol and water consumption were normalized for evaporation[70]. Briefly, the total volume of the liquid evaporated was calculated by averaging 2-days of evaporation from the four control cages without mice. Then, the volume of water or ethanol that evaporated was subtracted from the total consumption of water or ethanol for each mouse.

## Data analysis

All data were processed using Microsoft Excel (2013; Redmond, WA), then compiled and analyzed with Prism 9.0 (GraphPad Software, San Diego, CA). All data are represented as mean ± SEM. All tests were two-tailed and statistical significance was set at $p < 0.05$. Detailed statistical tests and data with exact $p$ values are listed in Supplementary Table 2.

## Reporting summary

Further information on research design is available in the Nature Portfolio Reporting Summary linked to this article.

# Data availability

The data generated in this study are provided in the source data file. Source data are provided with this paper.

# Code availability

All code used in this manuscript is available at https://doi.org/10.5281/zenodo.7908668.

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

## Acknowledgements

The authors thank all the laboratory members for their helpful discussion and comments. Figures were created with BioRender.com. This research was supported by the Samuel C. Johnson for Genomics of Addiction Program at Mayo Clinic, the Ulm Foundation, and the National Institute of Health (AA029258, AA028968, AA027773, AG072898) to D.S.C. This work was supported by the National Research Foundation of Korea (NRF) grant funded by the Korea government (MSIT) (NRF-2019M3E5D2A01066267, Development of meta-cognitive AI for rapid learning), Institute of Information & communications Technology Planning & Evaluation (IITP) grant funded by the Korea government (MSIT) (No. RS-2023-00233251, System3 reinforcement learning with high-level brain functions), and Institute for Information & Communications Technology Planning & Evaluation (IITP) grant funded by the Korea government (No. 2017-0-00451) to S.W.L.

## Author contributions

M.B., S.K., and D.S.C. thought of the study. M.B., S.K., and S.H. performed all behavioral and tracing experiments. M.B., L.P., and H.E. collected, processed, and imaged tissue for histology. M.S., M.A.Y., and S.W.L. sorted and analyzed Ca2+ signaling data and created SVM models. M.B. and D.S.C. wrote the manuscript. All authors reviewed and edited the manuscript.

## Competing interests

D.S.C. is a scientific advisory board member to Peptron Inc. Peptron had no role in the preparation, review, or approval of the manuscript; nor the decision to submit the manuscript for publication. The remaining authors declare no competing interests.
