## [Peer Review File · Nature Communications]

External globus pallidus input to the dorsal striatum regulates habitual seeking behavior in male miceREVIEWER COMMENTS

Reviewer #1 (Remarks to the Author):

This is a relatively straightforward, well-written novel and intriguing study, indicating a role for arkypallidal neurons in the performance of habitual behaviors.

I have some suggestions, which may further improve the study:

1. The indication of arky neurons projecting primarily to the DLS is convincing. There are, however, some small issues with the figures. In the intro, the author cite publications stating that 2/3 of GPe neurons are prototypic, while 1/4 are arkypallidal. However, the staining for FoxP2 and PV that the authors show in Figure 2 does not appear to be consistent with these proportions (the image illustrates more FoxP2 than PV neurons). Furthermore, the venn diagrams and quantifications provided in the text appear to be more of guesstimates than quantification with interindividual variance. Could the authors quantify their data and explain any discrepancy with literature, if it arises?

2. It is unclear how the averages & variance of the photometry data was calculated. Is this data from averaging all trials within each mouse, and the SEM represents the variance across mice?

3. Overall, if I understand correctly, the quantification of the photometry data is DF/F , normalized to the 3sec prior to time alignment. This may introduce changes to the signal that are dependent on recent history, and if the mice are active in the task (as is expected), the current signal may be a result of a prior action. Perhaps it would be better to analyze the data by performing df/f over the whole session (taking F0 as the bottom 1% of the signal), followed by a z-score of the whole session? This should allow for a better comparison across mice.

4. If I understand correctly, the SVM provides $\pm 60\%$ classification accuracy on a binary decision (i.e. chance is 50%). Is this the case?

5. Finally, the Caspase ablation experiment provides a strong indication for a role of arky neurons in restricting the transition from goal-directed to habitual behavior, however, the 'converse' experiment with Gq activation, did not appear to provide consistent observations. In this case, no effect of Gq activation on devaluation is observed, but rather a global effect on performance is observed. The authors then include experiments with D1/D2 pharmacology, and claim that D1 agonism rescues the

behavioral deficit, but this experiment lacks the important reference of the impact of the D1 agonist on its own (in the absence of Gq activation). Furthermore, goal-directed performance is not addressed in this paradigm. All-in-all, as it currently stands, I find it hard to interpret the data presented in figure 7, especially due to the lack of experimental controls and references. I do not find that it supports the string statements in the abstract ("Conversely, chemogenetic activation reduced habitual [AC1] seeking-behaviors, which was blocked by systemic D1R agonism") and the discussion ("chemogenetic activation led to reduced habitual reward-seeking, which was prevented by D1R agonism").

I suggest that the authors either try to enhance the experimental support for their statements, or modify their interpretation in a way that is more directly aligned with the observations.

Reviewer #2 (Remarks to the Author):

This study by Baker et al. evaluated the regulation of habitual reward-seeking behavior in mice via the external globus pallidus-dorsal striatum axis, using behavioral analysis coupled with fiber photometry, genetic ablation, and chemogenetic activation of the external globus pallidus arypallidal neurons projecting to the dorsal striatum, as well as the use of pharmacological blocking by d1/d2 receptors antagonists. Findings show that the external globus pallidus-dorsal striatum axis is involved in the habitual reward-seeking behavior that is relevant to compulsive disorders such as drug addiction. Overall, the paper is well-written, and the findings are presented in a manner that is easily understood. However, some points should be clarified and presented, particularly in the analysis or the conduct of behavioral analysis, as this is crucial in interpreting the molecular findings. These concerns have been listed below, which may contribute to improving the manuscript.

1. The authors state that during the operant conditioning tests, mice were placed on a restricted-food schedule to maintain 85% of their free-feeding weight. It would be helpful to show their body weights during the test period.
2. The authors should present the actual nose pokes during each session as a supplementary file.
3. Did the authors only use an FR1 schedule throughout the test? Have the authors tested an escalating reinforcement schedule?
4. How did the authors conclude the 30 minutes scheduled in RR or RI? This is too short for an experiment, particularly in the operant chamber, as rodents normally habituate for at least 10 minutes upon placement into the chamber. Also, are the mice producing 60 nose pokes during this duration?
5. Were the mice habituated in the open field and rotarod before the test?
6. Authors should include the movement duration of mice during the 60 min OFT and their visits towards the peripheral side of the box; this, along with the center visits, will confirm the absence of anxiety in mice for both groups.

7. The number of replicates is not well described.

Reviewer #3 (Remarks to the Author):

Baker and coworkers provide evidence that the activity of neurons that project from globus pallidus external segment to striatum influences the relative balance of goal directed and habitual operant nose-poke learning in mice. The authors use a combination of calcium measurements with fiber photometry, ablation of pallidostriatal projecting neurons, and DREADD-based modulation of these neurons to provide evidence that arypallidal neurons projecting to dorsolateral striatum may constrain habitual operant behavior. The findings are intriguing and the subject is certainly timely and suggests possible new roles for arypallidal neurons. The calcium measurements show clear differences between the RR and RI training conditions that generally support the authors' conclusions. However, there are concerns about several aspects of the study, including lack of quantification of data supporting some of the main conclusions, unclear presentation of the vector support machine analysis, and interpretation of the DREADD experiment. Many of the methods are described inadequately. Overall, this is an interesting and novel study, but there are doubts about some of the points.

Major Comments:

1. Without quantification of the data in figure 1 it is difficult to evaluate the authors' conclusions. It is not even clear how many animals were examined in this experiment. The authors need to calculate the projection densities in the different striatal subregions in a reasonable sample of animals.
2. Figure 2 contains more analysis, but it is still unclear how many total neurons were counted in the pie charts. Also, a chi square or Fisher's exact test should be used to compare the proportions of the different neuronal subtypes in panel c.
3. Quantification of the proportion of retrogradely-labelled and/or lesioned neurons that are FoxP2-positive is also needed for the experiments in figures 6 and 7. It appears that many, maybe even the majority, of retro-labeled neurons do not express FoxP2, but without quantification this is unclear. If many of the neurons that are lesioned or modulated by DREADDs are not FoxP2-expressing, the interpretation of these data may not be easily reconciled with the more specific calcium measurements in figure

4. The DREADD experiment is difficult to interpret. The rationale for examining effects of dopamine ligands is not compelling. The hM3Dq activation seems to produce a general decrease in behaviors. Since devaluation was not present in any group, even with the dopamine ligand treatments, it is difficult to conclude that there is any change in habitual responding. The conclusions stated on lines 290-292 are thus not supported by the data. In this context it is unclear why this experiment was only performed with the RI schedule, in contrast to the previous experiments. Floor effects in the C21 and raclopride group may also confound the interpretation of the devaluation test.

5. The SVM data are very difficult to understand for this reviewer, and likely for all readers not familiar with the approach. The data presentation in figure 5 appears to be standard for this type of analysis, but the authors need to provide a more detailed explanation of how the ROC curve data indicate how well the algorithm is classifying the data. Likewise, the accuracy, sensitivity and specificity values and their meanings need to be explained more clearly in the methods and results sections. The methods indicate that the classification used information from all trials from the RR and RI tasks. Presumably this was all training trials, so the question appears to be if signaling during training (as opposed to devaluation testing) predicts behavior, is that correct? Would results differ if signals during the devaluation test were used? The methods also appear to indicate that the SVM was "instructed" using a subset of the data that were then used for the prediction/classification analysis. If so, it would be informative to see how the SVM performed with data from a new cohort of mice. Ultimately it is not clear that this analysis adds much to the study. The difference in calcium signals over the different training regimens and valued/devalued conditions seems to support the overall conclusion without need for this additional and rather opaque approach.

6. The description of fiber photometry methods is too limited, even if the authors have previously published using this technique. They should at least indicate which system was used, if there is an isosbestic excitation wavelength control, and if so how any corrections using this channel and for photobleaching were done. If the short methods description is due to work limitations, a fuller description can always be included in the supplement.

Minor Comments:

i. The authors use the term habitual reward seeking several times in the manuscript. However, by definition habitual actions are not driven by reward, but rather by context and reinforcement history. Habitual responding is probably a better term.

ii. What antibodies were used? The authors need to provide the species, catalog number and dilution for each primary and secondary antibody. More information about the confocal imaging (e.g. lasers and filters used) is also required.

iii. Figures 1 and 2 might be combined, as they provide similar information. However, if additional panels are needed for quantification of data from the images, it might be best to keep them separate.

iv. Figure 3 starts with the photometry methods, but the rest of the figure is focused on behavior. It might be best to put panels a and b into figure 4.

v. Figure 3d is probably not necessary as a main figure component, as this type of data have been shown many times in the past.

vi. Lines 345-345, it's probably not a good idea to suggest that sucrose is an "addictive drug".

vii. Line 73, strictly speaking fiber photometry is not imaging as no image is generated.

viii. Line 421, what was the total volume of virus injected?

ix. There are several typographical errors and instances of awkward grammar that should be corrected. For example, the authors write d or iMSN's several places where they mean the term to be plural not possessive (e.g. line 293). Also, on lines 396-397 the last phrase could be construed to mean that reward was delivered continuously for 30 minutes.

Reviewer #1 (Remarks to the Author):

This is a relatively straightforward, well-written novel and intriguing study, indicating a role for arkypallidal neurons in the performance of habitual behaviors.

I have some suggestions, which may further improve the study:

1. The indication of arky neurons projecting primarily to the DLS is convincing. There are, however, some small issues with the figures. In the intro, the author cite publications stating that 2/3 of GPe neurons are prototypic, while 1/4 are arkypallidal. However, the staining for FoxP2 and PV that the authors show in Figure 2 does not appear to be consistent with these proportions (the image illustrates more FoxP2 than PV neurons). Furthermore, the venn diagrams and quantifications provided in the text appear to be more of guesstimates than quantification with interindividual variance. Could the authors quantify their data and explain any discrepancy with literature, if it arises?

Response: Thank you for pointing out the discrepancy, we have included comprehensive quantification (including mouse #, slice #, and variance) of arky cells. In coordination with reviewer 3's comments, we have combined Figures 1 and 2 into the new Figure 1, focusing on the main finding that GPe arkypallidal cells primarily project to the DLS compared to the DMS (see new Figure 1).

Additionally, in coordination with reviewer 3's comment asking, "what proportion of retrogradely labeled cell's are FOXP2", we have clarified the FOXP2 vs PV expression levels in the new Supplementary Figure 1. Our quantification and new representative images of overall FOXP2+ and PV+ cell numbers align with the literature mentioned. Of

note, we mentioned in the text that our study is “projection-based” and not “cell type-specific”, meaning we would expect that not all GPe→DLS cells express FOXP2.

2. It is unclear how the averages & variance of the photometry data was calculated. Is this data from averaging all trials within each mouse, and the SEM represents the variance across mice?

Response: That is correct, the presented data is the average from all the trials within each mouse, and the SEM represents the variance between mice.

3. Overall, if I understand correctly, the quantification of the photometry data is $\Delta F/F$, normalized to the 3sec prior to time alignment. This may introduce changes to the signal that are dependent on recent history, and if the mice are active in the task (as is expected), the current signal may be a result of a prior action. Perhaps it would be better to analyze the data by performing df/f over the whole session (taking F0 as the bottom 1% of the signal), followed by a z-score of the whole session? This should allow for a better comparison across mice.

Response: For the fiber photometry experiments, we wanted to assess relative changes in calcium signaling directly before, during, and after specific behavioral events. As the reviewer points out, our calculation for baseline fluorescence should be “F” and not “F0”. For our “F” calculation as the average of 3 seconds prior to time alignment, this allows for a direct comparison of the relative change in calcium signaling from moment to moment as each behavioral event occurs. A ‘sliding average’ in “F” metric is optimal for assessing relative changes between individuals and can provide a cleaner baseline. We agree that neural data around the behavior of interest might have been influenced by mice’s recent action, however it would be expected to be much smaller than the acute effect of behavior on the signal, and still allow for accurate comparisons between individuals. Especially as the behavioral events are aligned in time.

Regarding Z-score – while it has been often used, we chose not to use it due to z-score erasing variance differences between the two experimental groups (RR and RI) from the data. The variance of dF/F for RR and RI were 0.0075 and 0.012, respectively, and significantly differed (t-test, $p < 0.001$).

4. If I understand correctly, the SVM provides $\pm 60\%$ classification accuracy on a binary decision (i.e. chance is 50%). Is this the case?

Response: This is correct. In conjunction with reviewer 3's comments, we have expanded on the rationale and interpretation of the SVM results to make them more straightforward. Additionally, we have changed the SVM figure to clarify that the measures are compared to the 50% random chance (see new Figure 3b).

5. Finally, the Caspase ablation experiment provides a strong indication for a role of arky neurons in restricting the transition from goal-directed to habitual behavior, however, the 'converse' experiment with Gq activation, did not appear to provide consistent observations. In this case, no effect of Gq activation on devaluation is observed, but rather a global effect on performance is observed.

Response: We agree with the reviewer that the caspase ablation and chemogenetic activation results are less consistent than we have presented. Thus, we have added additional context to clarify the differences between the findings of two methods and the possible meaning in the discussion. In addition, as discussed in the following response, we have added additional open field and devaluation data to clarify our findings further. Specifically, while caspase ablation led to a transition to habitual behavior, Gq activation led to a global reduction in seeking-behaviors, without affecting locomotor behaviors (see new Supplementary Figures 7, 8, 10, and 11).

The authors than include experiments with D1/D2 pharmacology, and claim that D1 agonism rescues the behavioral deficit, but this experiment lacks the important reference of the impact of the D1 agonist on its own (in the absence of Gq activation).

Response: The reviewer brings up an important point. We mention in the discussion the limitations of these findings due to the systemic nature of the injections, as well as the future work needed to clarify the circuit-level effects of arky activation on D1R/D2R systems in the DLS. First, we performed a new experiment and added calcium imaging data confirming that C21 increases the calcium signal of DREADD expressing GPe arky pallidal neurons (see new Figure 5d).

Additionally, we have added open-field data for all the pharmacological injection groups. We found no significant differences in spontaneous locomotive behavior following C21 alone, nor C21+D1R agonist groups. This suggests that DREADD activation reduced seeking levels in the devaluation task without suppressing overall locomotor behavior. This supports our claims that the observed effect of C21 is more specific to reward-seeking than global motor function (see new Supplementary Figure 8).

Lastly, as requested by the reviewer, we carried out a new experiment and added additional devaluation data comparing the effects of the D1R agonist alone. As displayed below, we did find some evidence of increased perseverative behavior

indicated by a significant increase in magazine entry behaviors. However, there was no increase in nose pokes or magazine duration (see new Supplementary Figure 10).

Furthermore, goal-directed performance is not addressed in this paradigm.

Response: We carried out new experiments and added additional data to compare the effects of saline vs C21 in RR-trained animals (see new Supplementary Figure 11). We mention that due to the caspase findings, we sought to specifically test the hypothesis that arky activation could reverse or reduce habitual-seeking behaviors. Given our findings of global reductions in seeking-behaviors, we agree assessing goal-directed performance may be warranted. Again, we found decreases in overall seeking behaviors without changes to the goal-directed reward-seeking.

All-in-all, as it currently stands, I find it hard to interpret the data presented in figure 7, especially due to the lack of experimental controls and references. I do not find that it supports the string statements in the abstract ("Conversely, chemogenetic activation reduced habitual [AC1] seeking-behaviors, which was blocked by systemic D1R agonism") and the discussion ("chemogenetic activation led to reduced habitual reward-seeking, which was prevented by D1R agonism". I suggest that the authors either try to enhance the experimental support for their statements, or modify their interpretation in a way that is more directly aligned with the observations.

Response: We thank the reviewer for the critical considerations for the chemogenetic experiments. As above, we have reworded our conclusions and interpretations to focus on global reductions in seeking (see new Supplementary Figure 13) and added additional data to clarify the role of chemogenetic activation of arky neurons (see new Supplementary Figure 11).

Reviewer #2 (Remarks to the Author):

This study by Baker et al. evaluated the regulation of habitual reward-seeking behavior in mice via the external globus pallidus-dorsal striatum axis, using behavioral analysis coupled with fiber photometry, genetic ablation, and chemogenetic activation of the external globus pallidus arkypallidal neurons projecting to the dorsal striatum, as well as the use of pharmacological blocking by d1/d2 receptors antagonists. Findings show that the external globus pallidus-dorsal striatum axis is involved in the habitual reward-seeking behavior that is relevant to compulsive disorders such as drug addiction. Overall, the paper is well-written, and the findings are presented in a manner that is easily understood. However, some points should be clarified and presented, particularly in the analysis or the conduct of behavioral analysis, as this is crucial in interpreting the molecular findings. These concerns have been listed below, which may contribute to improving the manuscript.

1. The authors state that during the operant conditioning tests, mice were placed on a restricted-food schedule to maintain 85% of their free-feeding weight. It would be helpful to show their body weights during the test period.

Response: We have added additional supplementary data (see new Supplementary Figures 2, 5, 9) to show the food restriction weights in the period leading up to training.

2. The authors should present the actual nose pokes during each session as a supplementary file.

Response: In addition to the nose poke rate graphs, we have included the raw nose poke values per session in supplementary data (see new Supplementary Figures 2, 5, 9).

3. Did the authors only use an FR1 schedule throughout the test? Have the authors tested an escalating reinforcement schedule?

Response: Following FR1 training, we escalate RI30 to RI120 to model habitual-seeking, and RR2 to RR20 to model goal-directed reward-seeking. We have not tested an escalating reinforcement schedule, but it could be interesting to use an escalating fixed ratio or progressive ratio for the future studies.

4. How did the authors conclude the 30 minutes scheduled in RR or RI? This is too short for an experiment, particularly in the operant chamber, as rodents normally habituate for at least 10 minutes upon placement into the chamber. Also, are the mice producing 60 nose pokes during this duration?

Response: For the initial FR1 training schedule, sessions lasted for 60 minutes or 60 rewards. While the reviewer brings up a valid concern of the RR/RI sessions being too short to receive the full 60 reinforcements, our data still show stable or escalating nose poke rates across random schedule training days and expected reward devaluation

results. We believe the reviewer's above point to include raw nose poke values will also be essential to address this concern. So, we have added raw nose pole values in the Supplementary Figures, 2d, 5c and 9c.

5. Were the mice habituated in the open field and rotarod before the test?

Response: Animals were habituated to the testing room in their home cage 1 hour prior to behavioral testing but were not placed in the open field chamber or on the rotor rod prior to testing. We added these extra experimental timeline details to the methods section.

6. Authors should include the movement duration of mice during the 60 min OFT and their visits towards the peripheral side of the box; this, along with the center visits, will confirm the absence of anxiety in mice for both groups.

Response: We agree that stationary time could also be useful for assessing anxiety-like behavior. We added these data and found that caspase ablation and chemogenetic activation did not alter the anxiety-like behaviors (see new Supplementary Figures 4, 7, 8 and see the response to reviewer 1, point 5).

7. The number of replicates is not well described.

Response: In the method section of the revised manuscript, we clarified the number of replicates. Briefly, fiber photometry data was completed with RI first ($n = 5$) followed by RR ($n = 5$). The caspase data was performed in two cohorts, RI control/casp3 first ($n = 10/\text{group}$), followed by RR control/casp3 ($n = 10/\text{group}$). Then the replicate using 10% ethanol/10% sucrose reward was completed in one cohort ($n = 8/\text{group}$). DREADD data was completed in two cohorts ($n = 9$, then $n = 8$, 17 total). Then the replicate using a 10% ethanol/10% sucrose reward was completed in a second cohort ($n = 8$).

Reviewer #3 (Remarks to the Author):

Baker and coworkers provide evidence that the activity of neurons that project from globus pallidus external segment to striatum influences the relative balance of goal directed and habitual operant nose-poke learning in mice. The authors use a combination of calcium measurements with fiber photometry, ablation of pallidostriatal projecting neurons, and DREADD-based modulation of these neurons to provide evidence that arypallidal neurons projecting to dorsolateral striatum may constrain habitual operant behavior. The findings are intriguing and the subject is certainly timely and suggests possible new roles for arypallidal neurons. The calcium measurements show clear differences between the RR and RI training conditions that generally support the authors' conclusions. However, there are concerns about several aspects of the study, including lack of quantification of data supporting some of the main conclusions, unclear presentation of the vector support machine analysis, and interpretation of the DREADD experiment. Many of the methods are described inadequately. Overall, this is an interesting and novel study, but there are doubts about some of the points.

Major Comments:

1. Without quantification of the data in figure 1 it is difficult to evaluate the authors' conclusions. It is not even clear how many animals were examined in this experiment. The authors need to calculate the projection densities in the different striatal subregions in a reasonable sample of animals.

Response: Agreeing with the reviewer, we clarified the number of animals and quantification (see new Figure 1 and new Supplementary Figure 1). This additional analysis clarified that GPe projections target the DLS and not the DMS would significantly improve from quantification of the projection density. In coordination with reviewer 1 and the comments below, we have combined Figures 1 and 2, and added a Supplementary Figure 1 for cell-type quantification (see above response to reviewer 1).

2. Figure 2 contains more analysis, but it is still unclear how many total neurons were counted in the pie charts. Also, a chi square or Fisher's exact test should be used to compare the proportions of the different neuronal subtypes in panel c.

Response: Similar to point 1, we have added more precise quantification of FOXP2+ vs PV+ expression in the GPe retrograde experiments, along with sample sizes and variation (see new Supplementary Figure 1).

3. Quantification of the proportion of retrogradely-labelled and/or lesioned neurons that are FoxP2-positive is also needed for the experiments in figures 6 and 7. It appears that many, maybe even the majority, of retro-labeled neurons do not express FoxP2, but without quantification this is unclear. If many of the neurons that are lesioned or modulated by DREADDs are not FoxP2-expressing, the interpretation of these data may not be easily reconciled with the more specific calcium measurements in figure

Response: We have included the proportion of retrogradely labeled cells expressing FOXP2 in the new Supplementary Figure 1. We note in the discussion that while all FOXP2 cells project back to the dorsal striatum, studies report that they only make up 60-75% of the pallido-striatal cells, consistent with our findings. We further clarify in the discussion that our experimental strategies are primarily projection-based and not cell-type based. Future studies are warranted with Foxp2-Cre or Npas1-Cre mouse lines for additional cell-type specificity.

4. The DREADD experiment is difficult to interpret. The rationale for examining effects of dopamine ligands is not compelling.

Response: We agree with reviewer 1 and 3's concerns about the interpretation of the DREADD experiments. As mentioned above, we have simplified our explanation of our findings and added additional context through Supplementary Figures 7-12.

The hM3Dq activation seems to produce a general decrease in behaviors. Since devaluation was not present in any group, even with the dopamine ligand treatments, it is difficult to conclude that there is any change in habitual responding. The conclusions stated on lines 290-292 are thus not supported by the data. In this context it is unclear why this experiment was only performed with the RI schedule, in contrast to the previous experiments. Floor effects in the C21 and raclopride group may also confound the interpretation of the devaluation test.

Response: Agreeing with the reviewer's comment, we have reworded our interpretation of the DREADD experiment to match the global reduction more closely in seeking behaviors (new Supplementary Figure 13 and discussion). Additionally, we have added open field data for C21 concentrations and the pharmacologic combinations to supplemental data showing that while global seeking reductions occurred, this was not apparent for spontaneous locomotor behaviors (see new Supplementary Figures 7, 8, 10, and 11). This suggests that the global reductions in seeking behaviors are mostly specific to the seeking behaviors.

5. The SVM data are very difficult to understand for this reviewer, and likely for all readers not familiar with the approach. The data presentation in figure 5 appears to be standard for this type of analysis, but the authors need to provide a more detailed explanation of how the ROC curve data indicate how well the algorithm is classifying the data. Likewise, the accuracy, sensitivity and specificity values and their meanings need to be explained more clearly in the methods and results sections. The methods indicate that the classification used information from all trials from the RR and RI tasks. Presumably this was all training trials, so the question appears to be if signaling during training (as opposed to devaluation testing) predicts behavior, is that correct? Would results differ if signals during the devaluation test were used? The methods also appear to indicate that the SVM was "instructed" using a subset of the data that were then used for the prediction/classification analysis. If so, it would be informative to see how the SVM performed with data from a new cohort of mice. Ultimately it is not clear that this analysis adds much to the study. The difference in calcium signals over the different training regimens and valued/devalued conditions seems to support the overall conclusion without need for this additional and rather opaque approach.

Response: We thank the reviewer for their suggestion to expand on the rationale, interpretation, and value of the SVM analysis. We agree that many readers may not have experience in this method and the need to explain in more detail. Therefore, similar to our response to reviewer 1's concerns, we have modified the SVM figure (see new Figure 3) and added additional text in the methods and discussion to further explain the rationale and interpretation.

6. The description of fiber photometry methods is too limited, even if the authors have previously published using this technique. They should at least indicate which system was used, if there is an isosbestic excitation wavelength control, and if so how any corrections using this channel and for photobleaching were done. If the short methods description is due to work limitations, a fuller description can always be included in the supplement.

Response: In conjunction with reviewer 1's comments, we have expanded on the fiber photometry methods section, including these additional details.

Minor Comments:

i. The authors use the term habitual reward seeking several times in the manuscript. However, by definition habitual actions are not driven by reward, but rather by context and reinforcement history. Habitual responding is probably a better term.

Response: We agree with the reviewer's point. We removed "reward" and changed to habitual seeking behavior. We think, our habitual behavior is also not completely automatic or autonomous responding representing stimulus-response (SR) contingency since we measured the valued (V) and devalued (DV) responses during extinction. Then, we determine the goal-directed when animals reduce the responses in DV. In contrast, habitual behavior is when when animals are unable to reduce responses in DV (no difference between V and DV). So, it is a matter of shifting the balance between goal-directed and habitual behavior. Conceptually, we agree that if a behavior is completely SR, habitual responding makes sense.

ii. What antibodies were used? The authors need to provide the species, catalog number and dilution for each primary and secondary antibody. More information about the confocal imaging (e.g. lasers and filters used) is also required.

Response: We added all the additional antibody and microscopy-related information in the methods section.

“We used 488 (eGFP), 568 (mCherry), and 405 (Alexa Fluor® 405) excitation wavelengths for IHC imaging. All antibodies were purchased from Abcam (Cambridge, UK), and included anti-FOXP2 (rabbit; ab16046; 1:500) and anti-parvalbumin (rabbit; ab11427; 1:500) for primary, and anti-rabbit Alexa Fluor® 405 (donkey; ab175651; 1:500).”

iii. Figures 1 and 2 might be combined, as they provide similar information. However, if additional panels are needed for quantification of data from the images, it might be best to keep them separate.

Response: We appreciate the reviewer’s comment. As suggested, we have combined the two figures showing essential panels and quantification. Then, we have a new Supplemental Figure for detailed images and quantification.

iv. Figure 3 starts with the photometry methods, but the rest of the figure is focused on behavior. It might be best to put panels a and b into figure 4.

Response: As suggested, we have combined Figures 3 and 4 to make this change and have moved unnecessary training data to the new Supplementary Figure 2.

v. Figure 3d is probably not necessary as a main figure component, as this type of data have been shown many times in the past.

Response: As mentioned above, we move all basic training data to the new Supplementary Figures.

vi. Lines 345-345, it’s probably not a good idea to suggest that sucrose is an “addictive drug”.

Response: Agreeing with the reviewer, we have changed the wording at this line.

vii. Line 73, strictly speaking fiber photometry is not imaging as no image is generated.

Response: We agree with the reviewer’s point and have changed it accordingly.

viii. Line 421, what was the total volume of virus injected?

Response: We thank the reviewer for pointing out this missed detail and have updated it to show 400 nl virus injections.

ix. There are several typographical errors and instances of awkward grammar that should be corrected. For example, the authors write d or iMSN’s several places where they mean the term to be plural not possessive (e.g. line 293). Also, on lines 396-397 the last phrase could be construed to mean that reward was delivered continuously for 30 minutes.

Response: We apologize for these errors. We have incorporated these changes into the manuscript.

We again thank the reviewers for their time and helpful feedback. We hope that the revised manuscript is acceptable for *Nature Communications*.

Sincerely yours,

Doo-Sup Choi, Ph.D.
Professor of Pharmacology and Psychiatry
Director of Samuel C. Johnson Genomics of Addiction Program
Mayo Clinic College of Medicine
200 First Street SW
Rochester, MN 55905
Phone: 507-284-5602
Email: choids@mayo.edu

REVIEWERS' COMMENTS

Reviewer #1 (Remarks to the Author):

The reviewers have answered my queries in a satisfying fashion.

Congratulations!

Reviewer #2 (Remarks to the Author):

No further questions for the authors, and I approve the publication of the paper.

Reviewer #3 (Remarks to the Author):

The manuscript by Baker and coworkers is greatly improved. The new data, analyses, information and conclusions added to the manuscript has now clarified the story. However, the description of the fiber photometry methods is still incomplete. It is unclear if the authors used a commercially available system (e.g TDT or neurophotometrics), or a custom-built system. If a commercial system, does this include an "isosbestic" control excitation channel. If so, how were the data from this channel used in the calculation of the final $\Delta F/F$ calculations. What type of photodetector was used, i.e. PMT, CMOS camera, other? This information was also not provided in the previous Kang et al. paper (reference 38). The main concern is data such as that in figure 5D (similar to that in the Kang et al. paper). The waveforms in the C21 condition have odd shapes, looking more like plateauing increases than the fast-rise, exponential decay observed for most in vivo calcium increases. It is unclear if this has something to do with the data acquisition or analysis.

We want to thank reviewer #1 and reviewer #2 for endorsing the publication of our paper. We also thank reviewer #3 for asking to clarify the data acquisition and analysis of fiber photometry data. We addressed reviewer #3's remaining questions and revise the method section accordingly.

Reviewer #3

The manuscript by Baker and coworkers is greatly improved. The new data, analyses, information and conclusions added to the manuscript has now clarified the story. However, the description of the fiber photometry methods is still incomplete.

Response: We appreciate that the reviewer acknowledges the strength of our research. We revised the method section to include all the detailed points below.

It is unclear if the authors used a commercially available system (e.g TDT or neurophotometrics), or a custom-built system.

Response: We used a commercially available single-channel fiber-photometry with a Cinelyzer system (Ver. 1.0.1, Plexon, Dallas, TX).

If a commercial system, does this include an “isosbestic” control excitation channel. If so, how were the data from this channel used in the calculation of the final $\Delta F/F$ calculations.

Response: The single-channel fiber-photometry system does not contain an isosbestic control excitation channel. Instead, to approximate non-calcium dependent events, including autofluorescence, optical fiber artifacts, and photobleaching¹⁻⁵, a time-dependent baseline signal was computed by taking the minimum value of averaged Ca^{2+} trace in the 3s preceding the time alignment and filtering fast oscillatory noise by applying an exponentially weighted window. We described the detailed calculations in the updated methods section.

What type of photodetector was used, i.e. PMT, CMOS camera, other? This information was also not provided in the previous Kang et al. paper (reference 38).

Response: We used the Plexon single-channel CCD-based system.

The main concern is data such as that in figure 5D (similar to that in the Kang et al. paper). The waveforms in the C21 condition have odd shapes, looking more like plateauing increases than the fast-rise, exponential decay observed for most in vivo calcium increases. It is unclear if this has something to do with the data acquisition or analysis.

Response: In Figure 5, we recorded the signal to evaluate the effects of tonic stimulation of the GPe neuronal activities with chemogenetic approaches. The elongated signals have been observed in the case of pharmacologically induced tonic

cellular responses or continued behavior-synchronized cellular activities⁶⁻⁸. Thus, our elongated signals reflect the pharmacological activation of the DREADDs, which induced the consistent tonic stimulation of the neurons in the GPe. Thus, our observation reflects the fast-rise and exponential decay value when behavior-synchronized cellular responses with the fiber-photometry system (Figure 2).

Updated Fiber Photometry Methods Section:

In vivo Ca²⁺ signal with fiber-photometry. We recorded the cellular Ca²⁺ transients in real-time *in vivo* using single-channel fiber-photometry with a CineLyzer system (Ver. 1.0.1, Plexon, Dallas, TX)^{2,9}. We implanted a fiber-optic cannula (200/240 μm diameter, 200 μm end fiber) into the GPe (AP -0.46 mm, ML +2.0 mm, DV -3.8 mm from bregma) of mice injected with the AAV retrogradely expressing GCaMP6s in the DLS. The implanted fiber was linked to a patch cord, and the light intensity at the fiber tip was 60 μW consistently. These output signals were projected onto a CCD camera-type photodetector by the same optical fiber, passed through a GFP filter, and collected at 30 frames per second. For analysis of the photometry data, in the CineLyzer system, the calculation of $\Delta F/F$ values was conducted through three different stages^{2,10} and exported to MATLAB. First, the raw fluorescence $F_{RAW}(t)$ was averaged to get $F_{AVG}(t)$ over a sliding time window (0.75 s). The baseline for a particular frame $F_{BASELINE}(t)$ was the minimum of the $F_{AVG}(t)$ in a time window (3 s) preceding this frame. Second, $\Delta F/F(t)$ was calculated by subtracting $F_{BASELINE}(t)$ from $F_{RAW}(t)$ and then dividing it by $F_{BASELINE}(t)$. Finally, $\Delta F/F(t)$ was smoothed with an exponentially weighted moving window, which is described by a time constant, τ (0.2 s), and a width, w (1 s). For operant conditioning, fiber photometry was performed during the last sessions of RR20 and RI120. For chemogenetic validation, the Ca²⁺ signal was recorded for 10 minutes following systemic saline or C21 injection. Fiber photometry data were completed with RI animals first ($n = 5$) followed by RR ($n = 5$).

Reference in the response:

- 1 Siciliano, C. A. & Tye, K. M. Leveraging calcium imaging to illuminate circuit dysfunction in addiction. *Alcohol* **74**, 47-63, doi:10.1016/j.alcohol.2018.05.013 (2019).
- 2 Mu, M. D. *et al.* A limbic circuitry involved in emotional stress-induced grooming. *Nat Commun* **11**, 2261, doi:10.1038/s41467-020-16203-x (2020).
- 3 Gunaydin, L. A. *et al.* Natural neural projection dynamics underlying social behavior. *Cell* **157**, 1535-1551, doi:10.1016/j.cell.2014.05.017 (2014).
- 4 Falkner, A. L., Grosenick, L., Davidson, T. J., Deisseroth, K. & Lin, D. Hypothalamic control of male aggression-seeking behavior. *Nat Neurosci* **19**, 596-604, doi:10.1038/nn.4264 (2016).
- 5 Dai, B. *et al.* Responses and functions of dopamine in nucleus accumbens core during social behaviors. *Cell Rep* **40**, 111246, doi:10.1016/j.celrep.2022.111246 (2022).
- 6 Oyarzabal, E. A. *et al.* Chemogenetic stimulation of tonic locus coeruleus activity strengthens the default mode network. *Sci Adv* **8**, eabm9898, doi:10.1126/sciadv.abm9898 (2022).

- 7 Cho, J. R. *et al.* Dorsal Raphe Dopamine Neurons Modulate Arousal and Promote Wakefulness by Salient Stimuli. *Neuron* **94**, 1205-1219 e1208, doi:10.1016/j.neuron.2017.05.020 (2017).
- 8 Gao, C. *et al.* Two genetically, anatomically and functionally distinct cell types segregate across anteroposterior axis of paraventricular thalamus. *Nat Neurosci* **23**, 217-228, doi:10.1038/s41593-019-0572-3 (2020).
- 9 Kang, S. *et al.* Activation of Astrocytes in the Dorsomedial Striatum Facilitates Transition From Habitual to Goal-Directed Reward-Seeking Behavior. *Biol Psychiatry* **88**, 797-808, doi:10.1016/j.biopsych.2020.04.023 (2020).
- 10 Jia, H., Rochefort, N. L., Chen, X. & Konnerth, A. In vivo two-photon imaging of sensory-evoked dendritic calcium signals in cortical neurons. *Nat Protoc* **6**, 28-35, doi:10.1038/nprot.2010.169 (2011).

We again thank the reviewers for their time and helpful feedback. We hope that the revised manuscript is acceptable for *Nature Communications*.

Sincerely yours,

Doo-Sup Choi, Ph.D.
Professor of Pharmacology and Psychiatry
Director of Samuel C. Johnson Genomics of Addiction Program
Mayo Clinic College of Medicine
200 First Street SW
Rochester, MN 55905
Phone: 507-284-5602
Email: choids@mayo.edu